# Asymmetric histone inheritance regulates olfactory stem cell fates during regeneration

Binbin Ma[1,2], Guanghui Yang [1,2], Jonathan Yao[2], Charles Wu[2], Jean Pinckney Vega[2], Gabriel Manske[3,4], Saher Sue Hammoud [4], Satrajit Sinha [5], Abhyudai Singh [6], Haiqing Zhao [2] ✉ & Xin Chen [1,2] ✉

The olfactory epithelium possesses an adult stem cell population, the horizontal basal cells (HBCs), to permit lifelong tissue regeneration. Here, we show that HBCs exhibit asymmetric inheritance of histone H4 but not H2A-H2B during olfactory epithelium regeneration in mice. Primary HBC cultures further revealed asymmetric histone inheritance for H3 and H3.3. Upon mitotic exit, asymmetric histone inheritance correlates with asynchronous transcription re-initiation and differential enrichment of p63, a key transcription factor for HBC cell fate. Disruption of asymmetric histone inheritance abolishes these asymmetric cellular features and attenuates olfactory epithelium regeneration and smell behavior recovery. Single-cell RNA sequencing of paired HBC daughters in culture further supports asymmetric multilineage cell fate priming. Together, these findings reveal asymmetric histone inheritance in a mammalian adult stem cell lineage and highlight its biological significance in neural tissue regeneration and animal behavior.

The olfactory epithelium (OE) in the nose is a specialized neural epithelium responsible for the sense of smell. OE's location in the respiratory pathway makes it susceptible to airborne biological and chemical insults. The OE, however, has a remarkable capacity to repair and regenerate upon injury. In fact, OE is the sole neural tissue capable of continuously regenerating both neuronal and non-neuronal cell types throughout life[1]. This regenerative capacity is due to the presence of two stem cell populations in the OE, the horizontal basal cells (HBCs)[2,3] and some subtypes of globose basal cells (GBCs)[4]. HBCs reside along the basement membrane, forming a single layer at the bottom of the OE. GBCs reside on top of HBCs and are often organized into multiple layers[2,5].

Under normal physiological conditions, HBCs remain quiescent. In response to acute injury, however, HBCs are rapidly activated, re-enter the cell cycle, and give rise to all OE cell types[3]. HBCs express the transcription factor p63, a master regulator of stem cell fate[6–9]. p63 maintains the quiescent state of HBCs but also triggers their activation upon injury through its dowregulation[10,11]. Conditional p63 knockout

mice show spontaneous HBC differentiation in uninjured OE and defective HBC self-renewal following injury[9], underscoring its essential roles in balancing HBC cell fate specification during homeostasis and activation during regeneration. In the past years, a series of signaling pathways have been shown to regulate HBC self-renewal and differentiation[9,12–15]. However, how epigenetic mechanisms regulate HBC cell fate specification and OE regeneration remains largely unexplored.

As the primary carriers of epigenetic information, histones are critical for regulating gene expression and determining cell fate[16,17]. The nucleosome, the fundamental unit of chromatin, consists of two copies each of the canonical core histones—H3, H4, H2A, and H2B[18–20]. In addition to their structural roles, histones can undergo extensive post-translational modifications, which are crucial for their biological functions[21]. The composition, density, and position of nucleosomes, as well as distinct histone modifications, significantly contribute to cell fate decisions, including self-renewal and differentiation during homeostasis, as well as reprogramming during

[1]Howard Hughes Medical Institute, Department of Biology, Johns Hopkins University, Baltimore, MD, USA. [2]Department of Biology, Johns Hopkins University, Baltimore, MD, USA. [3]Cellular and Molecular Biology Graduate Program, University of Michigan, Ann Arbor, MI, USA. [4]Department of Human Genetics, University of Michigan, Ann Arbor, MI, USA. [5]Department of Biochemistry, Jacobs School of Medicine and Biomedical Sciences, SUNY at Buffalo, Buffalo, NY, USA. [6]Electrical & Computer Engineering, University of Delaware, Newark, DE, USA. ✉e-mail: hzhao@jhu.edu; xchen32@jhu.edu

tissue regeneration[16,17,22]. Previous studies have shown that histones H3 and H4 are inherited asymmetrically in germline stem cells[23–26] and intestinal stem cells[27] in *Drosophila* and in Wnt3a-induced mouse embryonic stem cells[28,29]. In *Drosophila* male germline stem cells, the asymmetric histone levels between sister chromatids are responsible for their differential condensation during mitosis and differential association of a key DNA replication re-initiation component Cdc6[30]. Despite these findings in *Drosophila* and mouse embryonic stem cells, it remains unknown whether mammalian adult stem cells possess asymmetric histone inheritance in vivo. And if yes, what is the biological significance in adult tissue?

In the current study, we combined in vivo stem cell tracking, primary HBC culture, and single-cell transcriptomics to characterize histone inheritance patterns during OE regeneration. Our results revealed that 30–40% of telophase HBCs exhibit asymmetric histone levels, accompanied by asynchronous transcription re-initiation and asymmetric p63 distribution in early regenerating OE. The transcriptomic analysis of paired HBC daughter cells revealed asymmetric multilineage cell fate priming after one round of division. We further showed that disruption of asymmetric histone inheritance led to a loss of both asynchronous transcription re-initiation and asymmetric p63 distribution in telophase HBCs. Notably, mice treated with nocodazole (NZ), a microtubule polymerization inhibitor that disrupts asymmetric histone inheritance, exhibited attenuated OE regeneration and impaired recovery of olfactory behaviors. Together, our results demonstrate that asymmetric histone inheritance plays a crucial role in regulating cell fate specification in mouse olfactory neural stem cells during regeneration.

## Results

### HBCs show asymmetric H4 inheritance during OE regeneration

Under standard laboratory conditions, HBCs are largely quiescent in adult mice[31]. The progenitor globose basal cells (GBCs) suffice olfactory sensory neuron turnover[32]. However, following acute injury, such as those causing the loss of sustentacular cells[15], HBCs become rapidly activated and divide into self-renewed HBCs and differentiating progenitor cells, which further give rise to all other OE cell types[3]. To investigate histone inheritance in HBCs during OE regeneration, we intraperitoneally injected 6- to 8-week-old adult mice with methimazole (MMZ), a drug commonly used to induce OE injury and regeneration in rodents[33–35]. A single MMZ injection (50 mg/Kg body weight) caused detachment of all OE cells, except HBCs, from the tissue 1 day after MMZ injection (Fig. 1A, Supplementary Fig. 1). Using Ki67 as a proliferation marker and p63 as an HBC marker, we found that HBCs undergo peak proliferation on Day 2 post-MMZ injection (Fig. 1B, left panel; Supplementary Fig. 1), indicating a rapid stem cell response during the early regeneration stage. Labeling with EdU (5-ethynyl 2´-deoxyuridine), a thymidine analog for S-phase cells, also revealed the same HBC proliferation dynamics (Fig. 1C). We thus chose Day 2 post-MMZ injection to examine histone inheritance in dividing HBCs. Of note, on Day 1 post-MMZ injection, quiescent HBCs become activated and re-enter the cell cycle through downregulation of p63[10,11] (Fig. 1B, right panel). Expression of p63 gradually resumes in proliferating HBCs, enabling differentiation and replenishment of the stem cell pool.

Among all histone types, we first focused on histone H4 because it has no known variants[36,37]; therefore, H4 levels faithfully reflect total nucleosome amount genome-wide. To visualize H4 distribution patterns, we used a Tet-inducible H4-tag transgenic mouse model, which overcomes the lack of effective immunohistochemistry reagents for H4. Mice carrying a tetracycline-responsive element (TRE3G)-driven H4-mScarlet (H4mS) were crossed with CAG-rtTA mice, allowing induction of H4-mScarlet expression upon doxycycline (DOX) injection in H4mS;rtTA progenies (Fig. 1D). We found that H4-mScarlet labeled all olfactory cell types, including p63+ HBCs (Fig. 1E). To assess

the relative expression levels of H4-mScarlet fusion protein compared with endogenous H4, we performed immunoblotting using OE tissues and found that the fusion protein accounts for ~6% of endogenous H4, indicating that endogenous H4 remains abundant and predominant in H4mS;rtTA OE cells (Fig. 1F). We further confirmed the dynamics of OE recovery post-MMZ injection between H4mS and H4mS;rtTA mice (Supplementary Fig. 2A). Compared to H4mS controls, H4mS;rtTA OE cells exhibited robust H4-mScarlet expression without significant differences in OE thickness or ratio of Ki67+ HBCs during early OE regeneration (Supplementary Fig. 2B, C). Thus, H4mS;rtTA mice allow visualization of H4 distribution patterns in vivo with only a minor increase in H4 amount, without compromising OE homeostasis or regeneration.

We found that, in early regenerating OE from H4mS;rtTA mice (Day 2 post-MMZ injection), H4-mScarlet was asymmetrically distributed in ~40% of p63-positive (p63 + ) telophase HBCs (Fig. 1G, H), using a stringent 1.5-fold threshold (Supplementary Fig. 2D). The remaining ~60% of p63+ telophase HBCs exhibited symmetric H4-mScarlet distribution. In contrast, p63-negative telophase cells predominantly (95%) showed symmetric H4 distribution (Fig. 1G, H). As a master regulator of HBC fates, p63 also showed asymmetric distribution in ~41% of telophase HBCs (Fig. 1I). Notably, when examining H4-mScarlet and p63 together in dividing HBCs, cells exhibiting asymmetric H4-mScarlet also showed asymmetric p63 distribution, with higher H4 levels correlating with greater p63 abundance between two sister chromatids (Fig. 1G, J). Next, we characterized the division angle of dividing HBCs and found that ~52% of asymmetrically dividing HBCs divided perpendicularly, whereas only ~1% of symmetrically dividing HBCs did so (Supplementary Fig. 3), suggesting that asymmetric cell fate specification may be influenced by spatially localized niche signals from the olfactory lamina propria.

In summary, during early OE regeneration, HBCs exhibit asymmetric histone H4 inheritance that is tightly associated with cell fate regulatory factors such as p63.

### HBCs exhibit symmetric H2A-H2B distribution

Canonical histone proteins are primarily synthesized and incorporated during DNA replication as an octamer structure consisting of two copies each of H3, H4, H2A, and H2B[18–20]. During nucleosome assembly, H3 and H4 are deposited as an intact $(H3\text{-}H4)_2$ tetramer, whereas H2A and H2B are incorporated as dimers[38]. To determine the specificity of asymmetric histone inheritance, we investigated the distribution of H2A–H2B dimers in the OE using nanobody-based detection (Supplementary Fig. 4A). Nanobody staining labeled H2A-H2B dimers in all OE cell types in uninjured OE, as well as in both p63+ HBCs and p63-negative early differentiated cells in injured OE on Day 2 post-MMZ injection (Supplementary Fig. 4B). In p63+ telophase HBCs, H2A-H2B levels were equal between the two sets of sister chromatids, regardless of p63 distribution (Supplementary Fig. 4C, D). Quantitative analysis of H2A-H2B and p63 ratios between sister chromatids confirmed a low correlation between them (Supplementary Fig. 4E). Similarly, in p63-negative telophase cells, H2A-H2B levels were consistently symmetric between sister chromatids (Supplementary Fig. 4C). Together, these results indicate that H2A-H2B is symmetrically inherited during HBC division and is not associated with cell fate regulatory factors during early OE regeneration.

### Cultured HBCs display asymmetric histone inheritance

To further investigate histone inheritance, we adopted primary HBC cultures, which have previously served as an ex vivo system to recapitulate in vivo HBC behavior and activity[39]. The culture system enables greater flexibility for manipulation, compatibility for histone immunostaining, and single-cell tracking. We further optimized the extracellular matrix coating to better modulate HBC activation (Fig. 2A). Specifically, Tropoelastin coating maintains HBCs in a

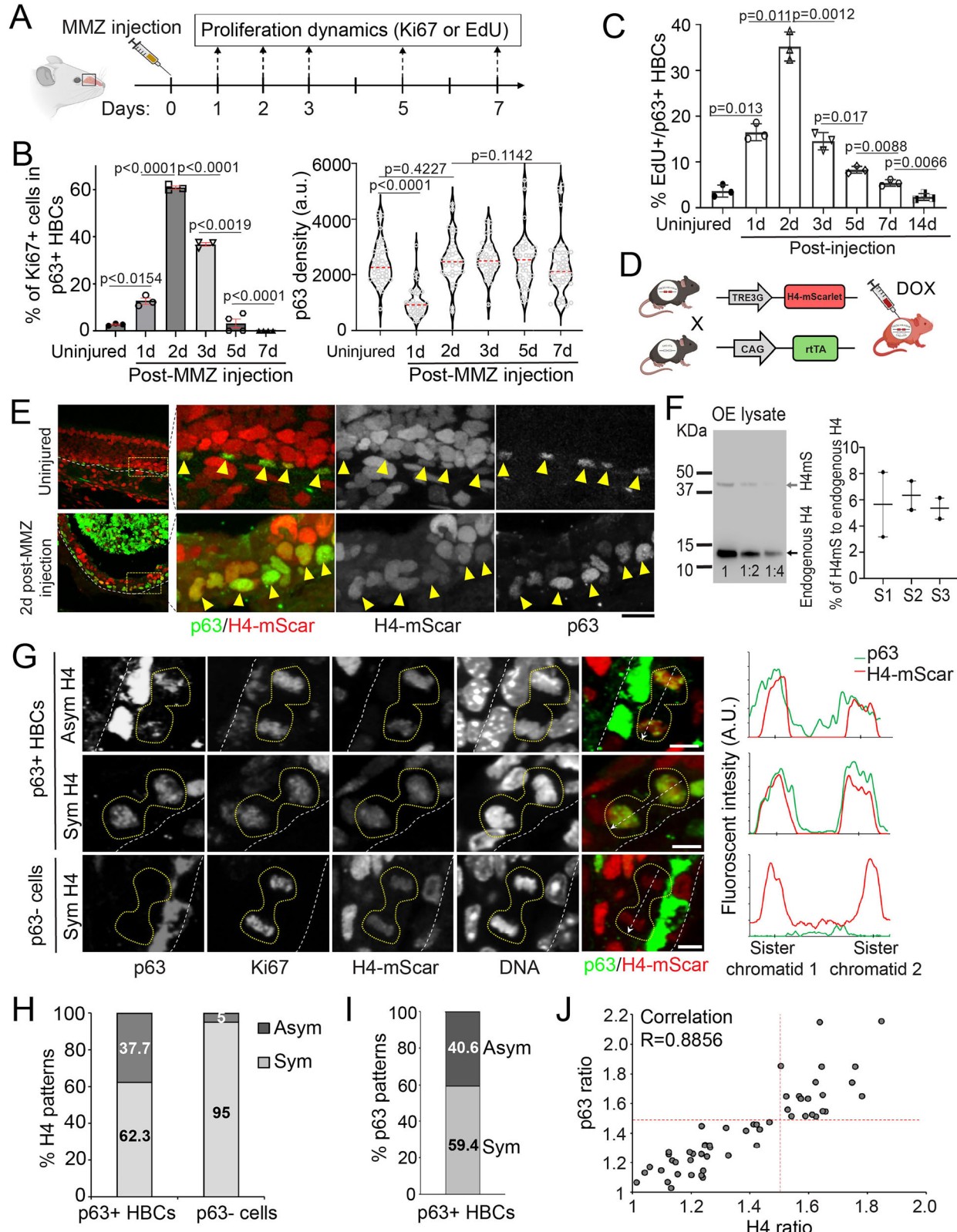

quiescent-like state, in which cultured HBCS form compact, small colonies with high p63 and low Ki67 levels (Fig. 2A, B). Fibronectin coating, on the other hand, induces colony expansion and cell flattening accompanied by p63 downregulation and a drastic upregulation of Ki67, indicating a transition from quiescence to an activated state (Fig. 2B). Consistently, the average cell cycle length decreased significantly from ~38 hours under Tropoelastin-coated conditions to

~14 hours under Fibronectin-coated conditions (Supplementary Fig. 5A–C), largely due to a significant shortening of the G1 phase. Mathematical modeling using the fractions of cultured HBCs in different cell cycle phases faithfully recapitulated the in vivo EdU labeling dynamics detected during OE regeneration (Supplementary Fig. 5D; compared with Fig. 1C). Thus, primary cultured HBCs under Tropoelastin- and Fibronectin-coated conditions reliably mimic in vivo stem

**Fig. 1 | Asymmetric histone H4 inheritance and asymmetric p63 distribution in dividing HBCs. A** Illustration of methimazole (MMZ) administration and olfactory epithelium (OE) sample collection for proliferation dynamics. **B** Left panel: Ratio of Ki67+ cells in p63+ HBCs during early OE regeneration. Right panel: Quantifications of p63 intensity in p63+ HBCs at different days during OE regeneration. a.u., arbitrary units. Statistical differences according to the two-sided Mann-Whitney test. $n$ = 3–4 biological replicates, and error bars represent mean$\pm$ SEM. **C** Quantifications of the percentage of EdU+ cells in p63+ HBCs at different days post-MMZ injection. $n$ = 3 biological replicates and error bars represent mean$\pm$ SEM. **D** Illustration of doxycycline-inducible H4-mScarlet expression in mice. **E** Uninjured and injured OE from H4mS;rtTA mice stained for p63. Arrowheads: p63+ HBCs. **F** Left panel: Immunoblot of H4mS;rtTA OE lysate on Day 2 post-MMZ injection with anti-H4 antibody (Sample 3 of Supplementary Fig. 1B). The three lanes represent the original OE lysate, 1:2 and 1:4 dilution of the original lysate, respectively. Top band, H4-mScarlet fusion protein, ~40 KD; Bottom band, endogenous H4 protein, ~11 KD. Right panel: Quantification of H4-mScarlet compared to endogenous H4 of three biological duplicates (Samples 1-3) in Supplementary Fig. 1B. Note: Due to the low signals of 1:4 dilution lane, the first two lanes were used for quantification. **G** H4-mScarlet distribution patterns in p63+ telophase HBCs from an OE section of H4mS;rtTA mice on Day 2 post-MMZ injection. Top, an HBC with asymmetric distributions of H4 and p63; Middle, an HBC with symmetric distributions of H4 and p63; Bottom, a p63-negative (p63-) cell with symmetric H4 distribution. Line plots on the right side show the distribution patterns of H4-mScarlet and p63. **H** Ratios of H4 distribution patterns in p63+ HBCs ($N$ = 53) and p63- cells ($N$ = 20) at telophase. **I** Ratios of asymmetric *versus* symmetric division modes in p63+ telophase HBCs ($N$ = 64). **J** Correlation of H4-mScarlet ratios and p63 ratios between two sister chromatids in p63+ telophase HBC. R = 0.8856 ($N$ = 53). Cutoff of asymmetry =1.5 based on the calculation in Supplementary Fig. 2D. Scale bars: 20 μm in (**E**), 5 μm in (**G**). The cartoon elements are created in BioRender. Chen, X. (2026) https://BioRender.com/r11witq. Source data are provided as a Source Data file.

---

cell properties, making them a suitable model for studying histone inheritance patterns.

Using histone antibody staining, we observed asymmetric H4 distribution in ~33% of telophase HBCs cultured on Fibronectin (Fig. 2C, C'), a frequency comparable to that detected in dividing HBCs in vivo (Fig. 1H). Asymmetric inheritance of histone H3 was also observed in ~35% of telophase HBCs (Fig. 2D, D'). Notably, both asymmetric H4 and H3 inheritance were accompanied by asymmetric p63 distribution, with the higher histone side correlating with higher p63 levels, mirroring the in vivo findings (Fig. 1G).

We next examined the inheritance pattern of the histone variant H3.3. In contrast to canonical histones, whose chromatin incorporation is DNA replication-dependent, H3.3 incorporation is independent of DNA replication but relies on transcription. We found that H3.3 displayed asymmetric inheritance in ~32% of telophase HBCs (Fig. 2E, E'). These cells also exhibited asymmetric p63 distribution, with higher H3.3 levels correlating with the p63-enriched side. In contrast, H2A-H2B predominantly displayed symmetric inheritance patterns (~95%) between sister chromatids, regardless of p63 distribution (Fig. 2F, F'), consistent with the in vivo results (Supplementary Fig. 4C, D).

Together, these results show that histones H4, H3, and H3.3 are asymmetrically inherited in dividing HBCs in primary cell culture, accompanied by asymmetric p63 distribution.

## (H3.3-H4)$_2$ tetramers regulate p63 chromosome rebinding

The asymmetric inheritance of H4, H3, and H3.3, together with asymmetric p63 distribution in dividing HBCs, prompted us to investigate how histone composition influences stem cell specification. We assessed the molecular association between different histones and p63 by quantifying the relative Pearson colocalization coefficients[27,40,41] in telophase HBCs, a stage when p63 rebinds to chromosomes as they decondense prior to entry into the subsequent interphase. Higher coefficients indicate greater colocalization, whereas lower coefficients indicate more separation between two signals.

We observed distinct degrees of colocalization between p63 and different histones (Fig. 2G). Notably, the histone variant H3.3, which is closely associated with active transcription and enriched in open chromatin regions, exhibited a high degree of colocalization with p63 in telophase HBCs (Fig. 2G, H). In contrast, canonical histones H3 and H2A-H2B exhibited significantly lower colocalization with p63 compared to H3.3, consistent with their enrichment in more closed chromatin regions (Fig. 2G, H). Interestingly, histone H4 exhibited a bimodal colocalization pattern with p63 (Fig. 2G, H): The higher degree of colocalization likely reflects the association of (H3.3-H4)$_2$ tetramers with p63, whereas the lower degree of colocalization reflects dissociation of (H3-H4)$_2$ from p63 (Fig. 2H). Therefore, these findings indicate that, in addition to (H3-H4)$_2$, (H3.3-H4)$_2$ tetramers act as

epigenetic information carriers that regulate p63 binding and transcriptional activity in HBCs.

## Asymmetrically dividing HBCs show asynchronous transcription

During mitosis, transcription is globally silenced as chromatin becomes highly condensed, restricting access of the transcription initiation complex and transcription factors. As dividing cells exit mitosis, transcription is reinitiated to reestablish transcriptomes that define the fates of the two daughter cells[42]. To assess how asymmetric histone inheritance is associated with transcriptional activity during HBC divisions, we examined Pol IIS2ph (RNA polymerase II phosphorylated at Serine 2, a marker of active transcription elongation)[43] in telophase HBCs on H4mS;rtTA OE sections (Fig. 3A). We observed asymmetric Pol IIS2ph patterns that aligned with asymmetric distribution of both histone H4 and p63 (Fig. 3A). This asymmetric distribution of phosphorylated Pol II suggests that HBCs undergo asynchronous transcription reinitiation between the two daughter nuclei during the M-to-G1 transition.

In primary cultured HBCs, we tracked the dynamic localization of Pol IIS2ph and another active transcription marker, Pol IIS5ph (RNA polymerase II phosphorylated at Serine 5, a marker of transcription initiation at promoter regions[44,45]). From prometaphase to anaphase, both Pol IIS2ph and Pol IIS5ph signals were diffusely distributed outside the chromosomes but rebound to decondensing chromosomes in late telophase (Fig. 3B, C; Supplementary Fig. 6A). Consistent with in vivo observations, both Pol IIS5ph and Pol IIS2ph were asymmetrically distributed in p63+ HBCs at telophase (Fig. 3D, E). These telophase cells also exhibited asymmetric p63 distribution, with quantification confirming a positive correlation between p63 ratios and Pol IIS5ph or Pol IIS2ph ratios (Fig. 3D', E').

We also performed immunostaining for unphosphorylated Pol II (Pan-Pol II) and found that it exhibited an overall symmetric distribution in telophase HBCs, regardless of whether p63 distribution was asymmetric or symmetric (Supplementary Fig. 6B–D). These results suggest that unphosphorylated RNA polymerase II is evenly distributed, whereas transcription is differentially reinitiated by Pol II S5ph and elongated by Pol II S2ph. Mathematical modeling further predicted an approximately 60% difference in RNA Pol II binding affinity between two sister chromatids carrying distinct cell fate information (Supplementary Fig. 6E, F). To directly monitor transcriptional activity, we used 5-ethynyluridine (EU) to label nascently transcribed RNA in cultured HBCs. Nascent RNA synthesis also exhibited asymmetric patterns, positively correlating with asymmetric p63 and Pol IIS2ph distribution in p63+ telophase HBCs (Fig. 3F, F').

Collectively, these results suggest that transcription is asynchronously reinitiated during mitotic exit and is regulated by nucleosome

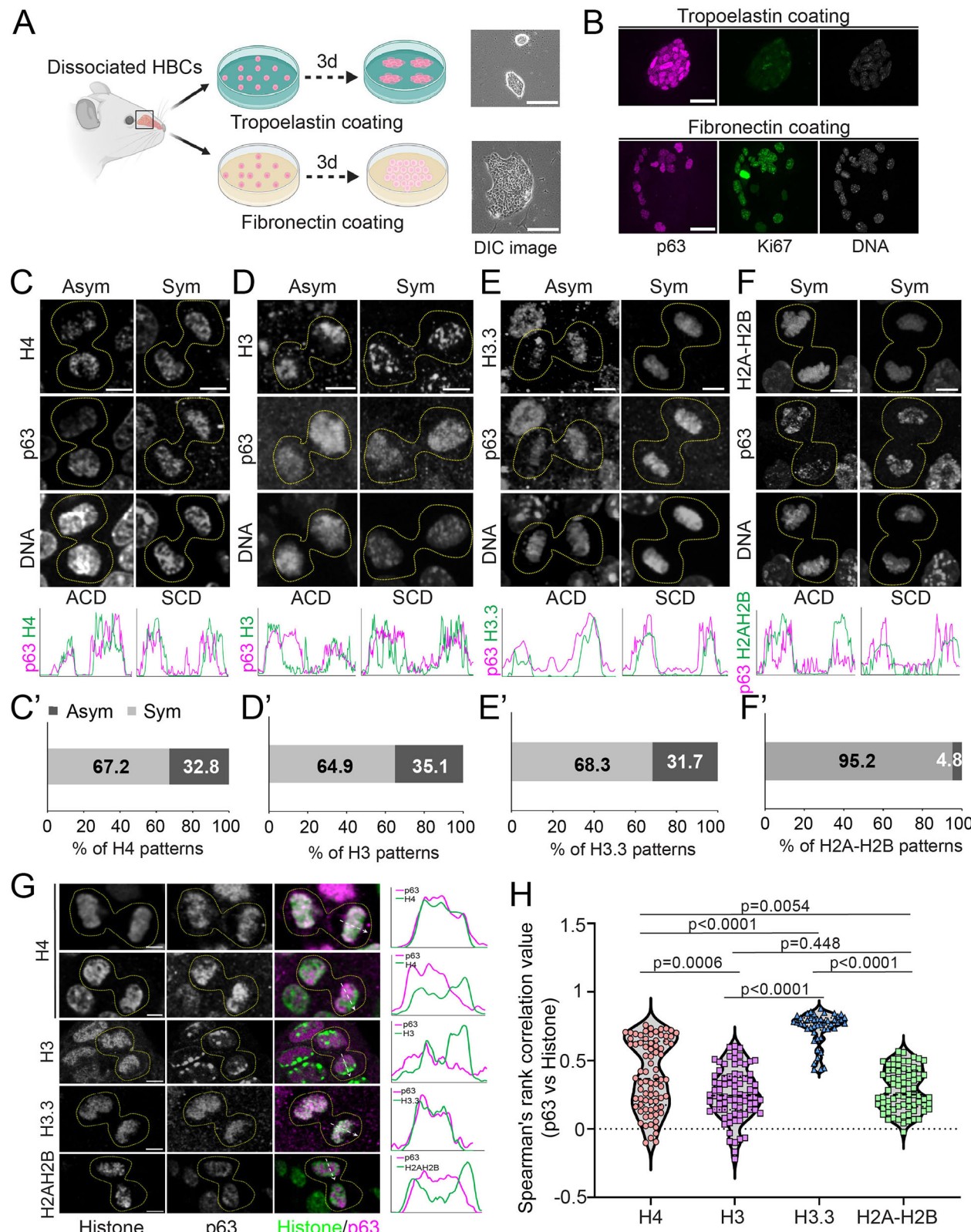

asymmetry between sister chromatids as chromatin decondenses, thereby likely contributing to distinct daughter cell fates.

## scRNA-seq of paired daughter HBCs reveals asymmetric fate priming

Single-cell RNA sequencing (scRNA-seq) has uncovered the heterogeneity of activated HBCs, with distinct subpopulations giving rise to

multiple OE lineages during post-injury repair[46]. To dissect the molecular features of asymmetrically dividing HBCs, we profiled single-cell transcriptomes of paired daughter HBCs in culture using the G&T-seq (Genome & Transcriptome-sequencing) method[47]. To facilitate isolation of paired daughter cells at single-cell resolution, we utilized a Tropoelastin-to-Fibronectin switch to promote effective HBC attachment and division (Fig. 4A). Live-cell imaging was then

**Fig. 2 | Asymmetric histone inheritance in primary cultured HBCs. A** Illustration of cultured HBCs with different extracellular matrix coating conditions. Tropoelastin maintains the dome morphology of HBCs, while Fibronectin promotes the flattened morphology of HBCs. **B** p63 and Ki67 staining of cultured HBCs on Tropoelastin-coated dish and Fibronectin-coated dish. **C** H4 distribution patterns in p63+ telophase HBCs. H4 is detected by anti-H4 antibody. Line plots show that the asymmetric H4 is associated with asymmetric p63 and that the symmetric H4 is associated with symmetric p63. **C′** Quantification showed the percentage of H4 patterns between sister chromatids of p63+ telophase HBCs (N = 64): Asymmetric H4 = 32.8%, Symmetric H4 = 67.2%. **D** H3 distribution patterns in p63+ telophase HBCs. H3 is detected by anti-H3 antibody. Line plots show that the asymmetric H3 is associated with asymmetric p63 and that the symmetric H3 is associated with symmetric p63. **D′** Quantification showed the percentage of H3 patterns between sister chromatids of p63+ telophase HBCs (N = 74): Asymmetric H3 = 35.1%, Symmetric H3 = 64.9%. **E** H3.3-mScarlet distribution patterns in p63+ telophase HBCs. Line plots show that the asymmetric H3.3 is associated with asymmetric p63 and that the symmetric H3.3 is associated with symmetric p63. **E′** Quantification showed the percentage of H3.3 patterns between sister chromatids of p63+ telophase HBCs

(N = 41): Asymmetric H3.3 = 31.7%, Symmetric H3.3 = 68.3%. **F** H2A-H2B distribution patterns in p63+ telophase HBCs. H2A-H2B is detected by an anti-H2A-H2B nanobody. Line plots show that the symmetric H2A-H2B is associated with either asymmetric or symmetric p63. **(F′)** Quantification showed the percentage of H2A-H2B patterns between sister chromatids of p63+ telophase HBCs (N = 42): Asymmetric H2A-H2B = 4.8%, Symmetric H2A-H2B = 95.2%. For panels in (**C**)−(**F**), Asym vs. Sym is based on histone patterns; ACD (asymmetric cell division) vs. SCD (symmetric cell division) is based on p63 patterns. **G** Images and line plots showing colocalization patterns of histone H4, H3, H3.3, and H2A-H2B, each with p63 in cultured telophase HBCs. **H** Colocalization of different histones with p63 in telophase HBCs. Average colocalization indexes are: H4, 0.39 ± 0.03; H3, 0.26 ± 0.02; H3.3, 0.74 ± 0.02; H2A-H2B, 0.28 ± 0.02. Values represent average ± SEM. Statistical differences according to the two-sided Mann-Whitney test. Cutoff of asymmetry = 1.5 based on the calculation in Supplementary Fig. 2D. Scale bars: 100 μm in (**A**, brightfield); 20 μm in (**B**, immunostaining); 5 μm in (**C**)−(**F**) and (**G**). The cartoons are created in BioRender. Chen, X. (2026) https://BioRender.com/18llh0f. Source data are provided as a Source Data file.

utilized to identify paired daughter cells following a single HBC division (Fig. 4B).

In total, 96 cells (48 pairs) were collected for scRNA-seq. After aligning all scRNA-seq reads to the reference genome, all 96 cells met the quality criteria. On average, $1.3 \times 10^4$ genes were detected per cell with ~2% of reads mapping to mitochondrial genes (Supplementary Fig. 7A). By comparing our results with published scRNA-seq datasets[12,46], we found that most cell pairs could be assigned to clusters corresponding to either renewed (less activated) or activated HBCs (Fig. 4C, Supplementary Fig. 7B). All 96 cells expressed HBC markers *Trp63* (encoding p63), *Krt14*, and *Krt5*, confirming their olfactory stem cell fate (Fig. 4D). Additionally, the majority of cells expressed wound-response and cell-cycle-associated genes, such as *Mki67* (encoding Ki67), *Rps6*, and *Krt6a*, consistent with HBC activation (Fig. 4E). In contrast, markers of other OE cell types were undetectable, including progenitor GBC markers (*Ascl1*, *Kit*, *Lgr5*), the sustentacular cell marker (*Il33*), and mature olfactory sensor neuron markers (*Gng13*, *Omp*) (Supplementary Fig. 7C, D). Together, these findings indicate that although the collected cells had acquired an activated state, they largely retained their stem cell identity.

Using cluster annotation from published datasets[12,46], we identified both symmetric and asymmetric daughter cell pairs (Fig. 4C, F). Among the 48 pairs, 22 consisted of two activated HBC daughters and 11 consisted of two renewed HBC daughters, indicative of symmetric cell fate priming. Notably, 15 pairs (~31%) exhibited asymmetric cell fate priming, including 9 Activated HBC: Renewed HBC pairs, 3 Activated HBC: GBCs/MV/INP pairs, 2 Activated HBC: Sustentacular cells pairs, and 1 Activated HBC: Olfactory sensory neuron (OSN) pair (Fig. 4F). These asymmetric pairs highlight multilineage cell fate priming that occurs immediately following a single round of stem cell division.

We examined p63 expressions in paired daughter cells and found that 35 pairs (72.9%) exhibited asymmetric cell fate specification based on p63 transcript levels (Supplementary Fig. 7E). This proportion is higher than the asymmetry observed at the protein level in telophase HBCs using immunostaining, likely reflecting increased transcriptional differences from mitosis to post-mitotic cells. We also examined the expression of the histone variant H3.3, whose transcripts are poly(A)-tailed and encoded by the *H3f3a* and *H3f3b* genes. We found that 15 pairs (~31%) exhibited asymmetric expression of either *H3f3a* or *H3f3b* transcripts (Fig. 4G). Using *H3.3* transcriptional asymmetry as a reference, we further observed that the daughter cell with higher *H3.3* levels preferentially expressed differentiation-associated genes, including wound-response genes (*Krt16*, *Yap1*), proliferation genes (*Mki67*, *Cdc20*), and lineage-specific genes (*Sox9*, *Ascl3*, *Il33*). In contrast, the daughter cell with lower *H3.3* levels tended to display transcriptional features consistent with either self-renewal (higher *p63*, *Trp53inp1*,

*Myc*) or differentiation (lower *Krt5*, *Krt14*, *Icam1*) state (Fig. 4H, Supplementary Fig. 7F).

Principal Component Analysis (PCA) revealed transcriptomic diversity and a population shift between renewed and activated HBCs among paired daughter cells (Fig. 4I). Genes enriched in activated HBCs were significantly associated with Gene Ontology (GO) terms related to 'cell division', 'cell cycle', 'mitotic cytokinesis', 'chromosome segregation', and 'mitotic sister chromatid segregation' (Supplementary Fig. 7G), suggesting that the transition from renewed to activated states may involve mitotic events. Differential gene expression analysis further identified 85 upregulated and 10 downregulated genes in activated HBCs (59 cells) compared with renewed HBCs (31 cells) (Fig. 4J). Among these differentially expressed genes, cell-cycle regulators (*Mki67*, *Cdk1*, *Cdc20*, *Ccnb1*), known HBC markers (*Trp63*, *Krt5*, *Krt14*), and potential HBC markers identified in this study (*Krt16*, *Krt23*) may serve as transcriptional markers of asymmetric cell division generating Activated HBC: Renewed HBC pairs (Fig. 4K, Supplementary Fig. 7H).

Notably, transcript levels of the histone methyltransferase SUV39H1, which catalyzes trimethylation of histone H3 lysine 9 (H3K9me3) to promote heterochromatin formation and gene silencing, were upregulated in activated HBCs (Fig. 4K). Consistently, an enriched association of the SUV39H1 protein was detectable on the p63-higher sister chromatid in telophase HBCs exhibiting asymmetric p63 distribution (Fig. 4L, Supplementary Fig. 7I). Because histone H3 is also asymmetrically inherited in association with p63 enrichment (Fig. 2D), the concurrent enrichment of SUV39H1 and its histone substrate H3 may synergistically promote heterochromatin formation. This increase in heterochromatin could, in turn, "lock in" an epigenomic state priming cells for terminal differentiation[48,49].

Together, paired daughter cell scRNA-seq revealed asymmetric multilineage cell fate priming, likely guided by asymmetric histone inheritance during early regeneration.

## Loss of histone asymmetry disrupts asynchronous transcription and asymmetric p63

In *Drosophila*, asymmetric microtubule dynamics facilitate preferential attachment of sister chromatids, enabling asymmetric histone inheritance and differential cell fate specification during male germline stem cell asymmetric divisions. Pharmacological disruption of microtubules with Nocodazole (NZ), a reversible microtubule depolymerizing drug, abolishes asymmetric microtubule dynamics, leading to randomized histone inheritance and impaired cell fate determination[23]. Here, we investigated microtubule dynamics in dividing HBCs by live-cell imaging using a tubulin dye, co-labeled by a p63 transcriptional reporter (*p63*-EGFP)[50]. We found that approximately 31% of dividing HBCs

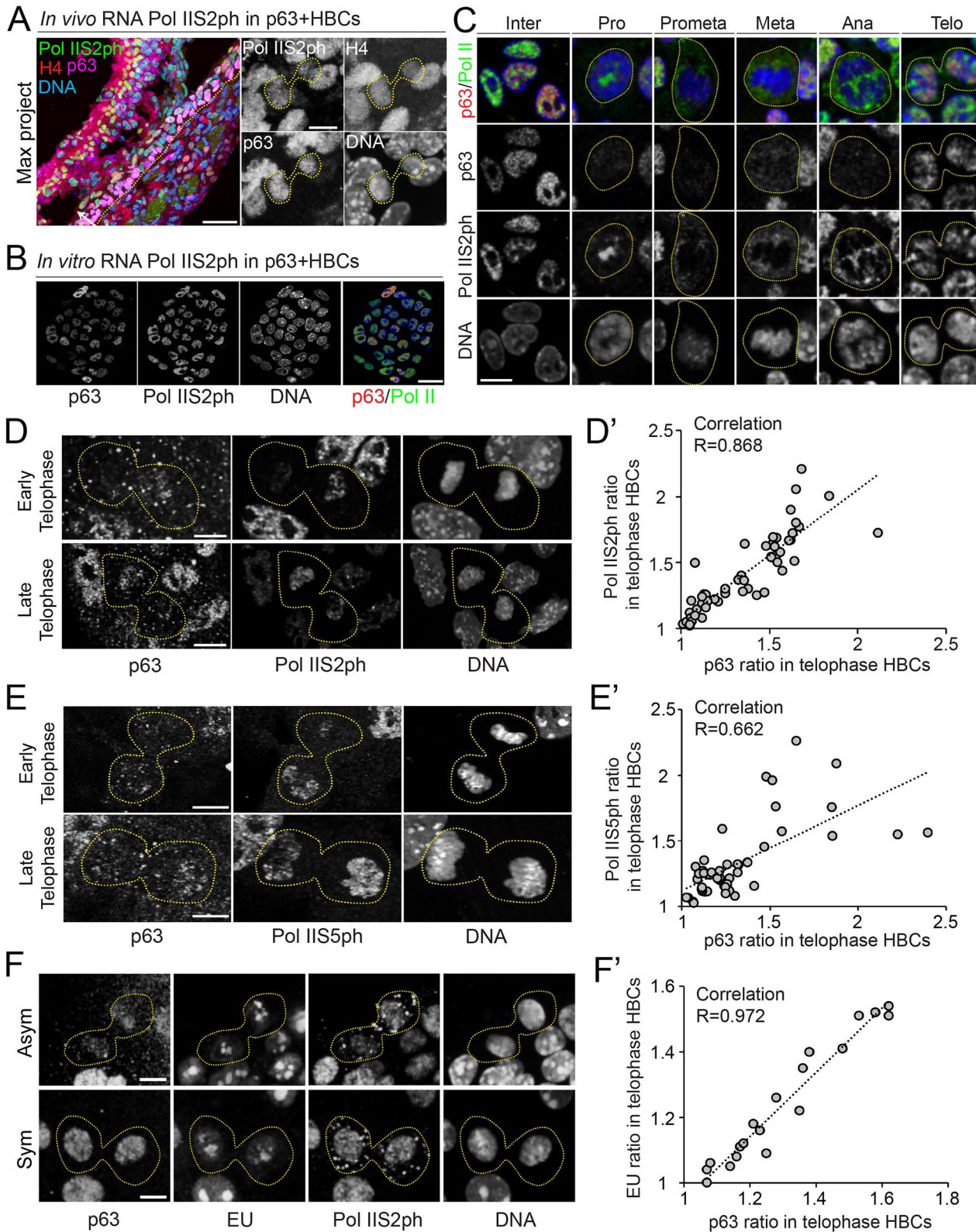

displayed transiently asymmetric microtubule dynamics, whereas the remaining dividing HBCs exhibited symmetric microtubule dynamics (Supplementary Fig. 8A).

We next assessed the effects of NZ treatment on histone inheritance in cultured HBCs (Supplementary Fig. 8B). Following a 16-hour NZ treatment, most HBCs were synchronized at prometaphase. Upon NZ washout, HBCs progressively re-entered mitosis, with the majority reaching telophase 30 minutes post-release

(Supplementary Fig. 8B). Under these conditions, live-cell imaging revealed that 100% of HBCs from H4mS;rtTA mice exhibited symmetric H4 segregation (Fig. 5A–C), in contrast to untreated controls, in which a substantial fraction (-30%) displayed asymmetric H4 segregation (Fig. 5B, C).

To investigate the consequences of disrupting asymmetric histone inheritance, we examined the distribution of Pol IIS2ph and p63 in NZ-treated HBCs. Immunostaining revealed a marked reduction in

**Fig. 3 | Asynchronous transcription re-initiation in asymmetrically dividing HBCs. A** Asymmetric RNA Pol IIS2ph distribution in p63+ telophase HBCs from injured OE (Day 2 post-MMZ injection). In the merged panel, the dotted line indicates the basal membrane, and the arrow points toward the epithelium surface. Notice that the telophase HBC that shows asymmetric RNA Pol IIS2ph (green) distribution also exhibits asymmetric distribution of H4 (red) and p63 (magenta). **B–D** Asymmetric RNA Pol IIS2ph distribution in telophase HBCs from primary culture. **B** RNA Pol IIS2ph staining in primary cultured HBCs. **C** Dynamics of RNA Pol IIS2ph distribution during interphase and mitosis. **D** Examples of asymmetric RNA

Pol IIS2ph distribution in early and late telophase HBCs. **D'**. Correlation of Pol IIS2ph ratios and p63 ratios between two sister chromatids. R = 0.868 ($N$ = 55). **E** Examples of Asymmetric Pol IIS5ph distribution in early and late telophase HBCs from primary culture. **E'**. Correlation of Pol IIS5ph ratios and p63 ratios between two sister chromatids. R = 0.662 ($N$ = 50). **F** Asymmetric and symmetric EU distribution in telophase HBCs from primary culture. **F'**. Correlation of EU ratios and p63 ratios between two sister chromatids. R = 0.972 ($N$ = 19). Scale bars: 50 μm in (**A**, section) and 5 μm in (**A**, telophase HBCs); 20 μm in (**B**); 10 μm in (**C**); 5 μm in (**D**)–(**F**). Source data are provided as a Source Data file.

asymmetric distribution of both Pol IIS2ph and p63 in telophase HBCs (Fig. 5D, E; Supplementary Fig. 8C).

In *Drosophila*, expression of histone H3 carrying a threonine-to-unphosphorylatable alanine at residue 3 (H3T3A) disrupts asymmetric H3 inheritance in male germline stem cells and leads to germ cell loss and germline tumors[26]. We introduced H3T3A in cultured HBCs derived from H3.3mS;rtTA mice via viral transfection (Fig. 5F) and found that H3T3A disrupted asymmetric histone inheritance in telophase HBCs (Fig. 5G, J), recapitulating its effects in *Drosophila* male germline stem cells. We next examined the distributions of Pol IIS2ph and Pol IIS5ph, markers of transcription reinitiation, as well as p63, and found that H3T3A expression also disrupted asynchronous transcription reinitiation (Fig. 5H–J) and asymmetric p63 distribution in telophase HBCs (Fig. 5G–J).

Together, these results suggest that disrupting microtubule dynamics by NZ treatment or directly perturbing chromatin through the H3T3A mutant histone abolishes asymmetric histone inheritance. Loss of asymmetric histone inheritance, in turn, leads to a loss of asynchronous transcription reinitiation and asymmetric cell fate specification in dividing HBCs.

### Loss of histone asymmetry impairs OE recovery after injury

To further investigate the in vivo consequences of losing asymmetric histone inheritance, we administered NZ one day after MMZ injection and accessed the OE regeneration process (Fig. 6A). During early regeneration, NZ-treated OE showed a significantly higher proportion of proliferating HBCs on Days 2 and 3 post-MMZ injection (Fig. 6B). In contrast to control (non-NZ treated) (Fig. 1I, Supplementary Fig. 3), in which ~40% of dividing HBCs exhibited asymmetric p63 distribution and perpendicular division angles on Day 2 post-MMZ injection, NZ-treated dividing HBCs predominantly exhibited symmetric p63 distribution and parallel division angles (Fig. 6C, D). These results indicate that NZ treatment disrupts the balance between asymmetric and symmetric cell fate specification and increases HBC proliferation during early OE regeneration.

To examine the consequences of early-stage disruption of asymmetric histone inheritance and HBC cell fate specification during OE regeneration, we further collected OE tissues on Days 15 and 28 post-MMZ injection (Fig. 6A, E). On Day 15, control mice showed substantial recovery of OE thickness and the number of olfactory marker protein (OMP)-positive mature OSNs (Fig. 6E, E'). In contrast, NZ-treated mice exhibited a significantly thinner OE with a thinner zone of the OMP-positive OSNs (Fig. 6E, E'), indicating attenuated regeneration. By Day 28, OE thickness and the OMP-positive OSN zone were further recovered in both non-NZ-treated control and NZ-treated mice; however, these parameters remained significantly reduced in NZ-treated mice (Fig. 6E, E'). Notably, this defect was dose dependent, with higher NZ doses resulting in a markedly thinner OE on Day 15 and a significantly reduced OMP-positive OSN zone on both Days 15 and 28 (Fig. 6E, E'). Overall, these results indicate that disruption of asymmetric histone inheritance in HBCs by NZ treatment leads to impaired OE regeneration.

To evaluate the functional impact of NZ treatment, we conducted a buried food-seeking behavioral test in NZ-treated mice. On Day 15 post-MMZ injury, 75% of control mice located the buried food pellet

within 10 minutes, whereas only 37% of NZ-treated mice succeeded (Fig. 6F). By Day 28, all control mice located the food in a timely manner, but 22% of NZ-treated mice still failed. An olfactory preference test[51] also showed that NZ-treated mice spent significantly less time exploring attractive odors than non-NZ-treated control mice on Day 15 post MMZ injection, whereas no significant difference was observed between the two groups on Day 28 (Supplementary Fig. 9). These olfactory behavioral deficits are consistent with the attenuated OE regeneration in NZ-treated mice.

Collectively, these in vivo results support that asymmetric histone inheritance in HBCs is required for the rapid regeneration of injured OE tissue and timely recovery of olfactory function (Fig. 6G).

## Discussion

In this study, we investigate histone inheritance and its role in cell fate specification in HBCs during regeneration of the mouse OE. We provide in vivo evidence of asymmetric histone inheritance in a mammalian adult stem cell population and show that this asymmetry occurs in a substantial fraction of dividing HBCs at the earliest stages of injury-induced regeneration. Asymmetric histone inheritance correlates with asynchronous transcription reinitiation and asymmetric distribution of p63, a master regulator of HBC fate, in telophase HBCs both in vivo and in primary culture. Disrupting histone inheritance asymmetry abolishes these transcriptional and p63 asymmetries, indicating their contributions to cell fate specification. Single-cell transcriptomic profiling of paired HBC daughter cells further reveals asymmetric multilineage cell fate priming. Finally, nocodazole treatment during early OE regeneration impairs olfactory sensory neuron regeneration and smell behavior recovery, underscoring the functional importance of asymmetric histone inheritance in tissue repair. Together, these findings highlight asymmetric histone inheritance as a biologically significant mechanism regulating stem cell function.

Asymmetric histone inheritance has been shown to regulate cell fate decisions in asymmetrically dividing *Drosophila* germline and intestinal stem cells[24,27]. Here, we report that asymmetric histone inheritance also regulates cell fate specification in a mammalian adult stem cell lineage, suggesting a conserved epigenetic mechanism in different adult stem cell systems across phyla. In olfactory HBCs, histone amount asymmetry is more pronounced for the canonical histones H3 and H4 than for H2A and H2B, consistent with previous findings in *Drosophila* adult stem cells[24,25]. Notably, the transcription-dependent histone variant H3.3[52,53], whose incorporation is replication-independent, also shows asymmetric inheritance. This finding suggests that (H3.3-H4)$_2$ tetramers may serve as key carriers of epigenetic information in HBCs. Because H4 lacks variants, total H4 levels likely reflect overall nucleosome density, whereas nucleosome composition and post-translational modifications (PTMs) provide locus-specific regulatory information. Indeed, H3.3-H4-containing nucleosomes are typically associated with active transcription, whereas H3-H4-containing nucleosomes can mark either active or repressive chromatin depending on their PTMs (e.g., active H3K4me3 and H3K36me3[54,55] versus repressive H3K9me3 and H4K20me3[56,57]). In this study, scRNA-seq of paired daughters revealed asymmetric multilineage priming, linking H3.3 transcript asymmetry to divergent differentiation trajectories (Fig. 4G, H). Additional sequencing-based

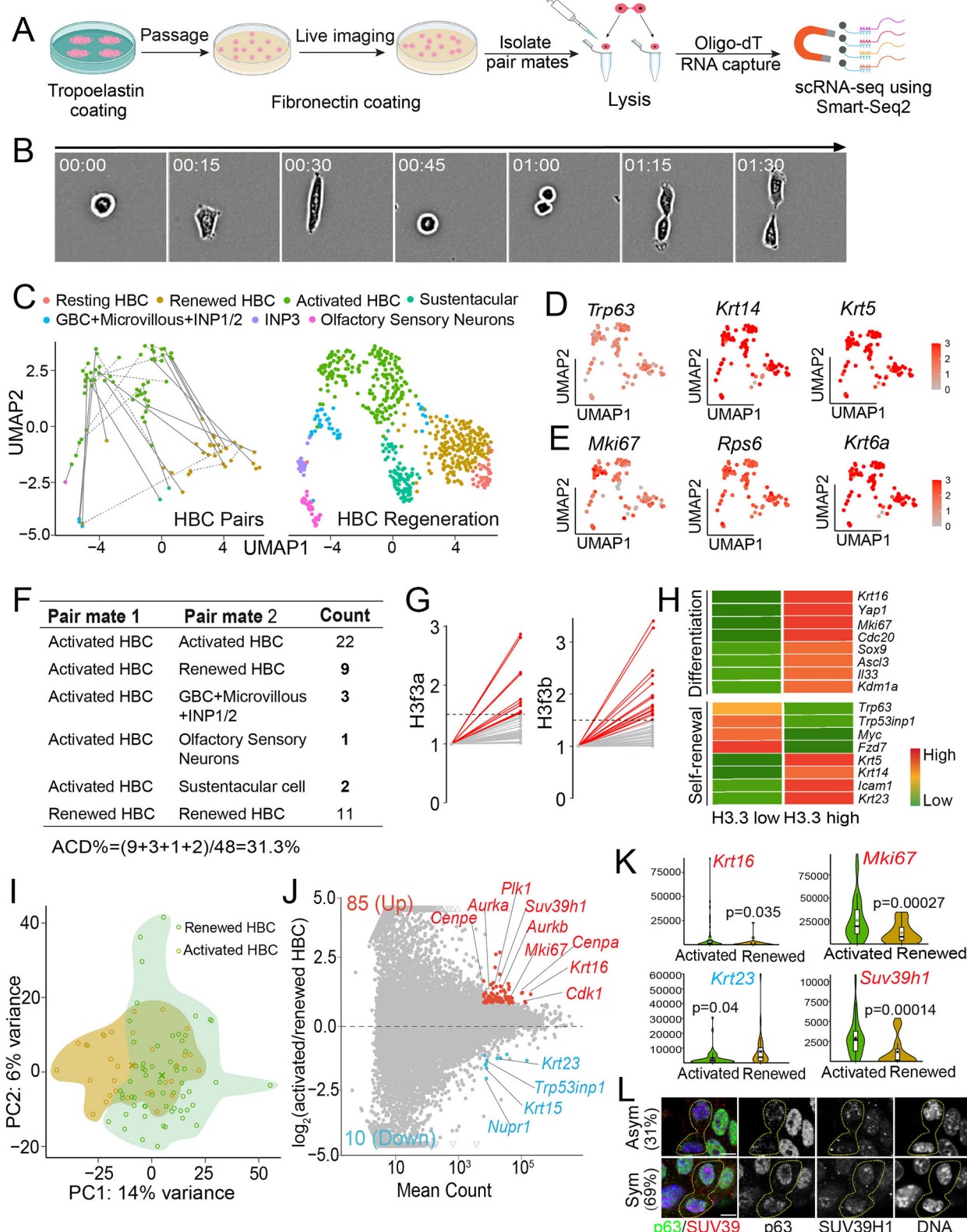

ACD%=(9+3+1+2)/48=31.3%

approaches will be needed to define how these chromatin features contribute to differential gene expression in the two daughter cells arising from an HBC division.

We further showed asymmetric enrichment of transcriptionally engaged phosphorylated RNA Polymerase II and the key fate determinant p63 with sister chromatids in telophase HBCs, indicating that chromatin differences underlie asynchronous transcription

reinitiation and transcription factor rebinding in daughter cells. Together with the asymmetric inheritance of H3.3, these findings reveal differential transcription reinitiation as a key biological consequence of asymmetric histone inheritance. Previous studies examined only old versus new H3.3 inheritance in *Drosophila* male germline stem cells[24,40] and did not assess total H3.3 levels or the transcriptional reactivation between sister chromatids. Our results extend previous

**Fig. 4 | Asymmetric multilineage cell fate priming of activated HBCs by paired daughter cell RNA sequencing. A** Scheme of paired-daughter cell preparation for scRNA-seq. **B** Live imaging of cultured individual HBC to track the division from a mother cell to two daughter cells. **C** Clustering of HBC paired cells (left panel) after integrating scRNA-seq data of the current study (GSE286046) into published dataset (GSE99251 and GSE95601). Solid lines indicate asymmetric pairs cross different clusters. Dotted lines indicate symmetric pairs in the same clusters. The right panel shows the clusters of OE cells derived from GSE99251 and GSE95601.
**D** Feature plots showing expression of HBC marker genes *Trp63*, *Krt14* and *Krt5* in collected single cells. **E** Feature plots showing expression of cell cycle and wound response genes *Mki67*, *Rps6* and *Krt6a* of collected single cells. **F** Statistics of paired information showing symmetric and asymmetric divisions in (**C**): 31.3% (15/48) of HBC pairs show asymmetric divisions. **G** Expression of H3.3 transcripts (*H3f3a/H3f3b*) in an individual pair. Red lines indicate the asymmetric pairs (N = 15). Grey lines indicate the symmetric pairs (N = 33). The asymmetric cutoff for *H3f3a* and *H3f3b* is 1.5. (**H**) Expression patterns of differentiation-associated genes (e.g., *Krt16*, *Sox9*, *Ascl3*, and *Il33*) (top panel) and self-renewal-associated genes (e.g.,

*Trp63*, *Fzd7*, *Krt14*, and *Krt5*) (bottom panel) in pairs with asymmetric H3.3. **I** PCA plot using scRNA-seq data of all renewed and activated HBCs. **J** MA-plot shows differentially expressed genes between renewed and activated HBCs. 85 genes (labeled in red) show higher expression in activated HBCs, 10 genes (labeled in blue) show lower expression in activated HBCs. **K** Differential expression of *Mki67*, *Suv39h1*, *Krt16*, and *Krt23* in activated and renewed HBC cells. Note that *Krt16* and *Krt23* are identified by the current analysis. Gene counts normalized by external ERCC spike-in are used in the violin and box plots. All box plots include the median and data between the 25th and 75th percentile, with the whiskers indicating lower and upper extremes after removing outliers. The black 'x' within each box shows the average within each group. A two-tailed t-test was used to calculate P-values.
**L** Asymmetric (31%) and symmetric (69%) SUV39H1 distribution in p63+ telophase HBCs (N = 29) from culture. p63 shows the same distribution pattern as SUV39H1. Scale bar: 5 μm in (**L**). The cartoon elements are created in BioRender. Chen, X. (2026) https://BioRender.com/yh44iq5 and Adobe Illustrator. Source data are provided as a Source Data file.

findings of asynchronous DNA replication reinitiation in daughter cells derived from *Drosophila* male germline stem cells[30]. Mechanistically, previous studies in *Drosophila* male germline stem cells have shown that both *trans*-acting asymmetric mitotic spindle activity and *cis*-acting H3T3 phosphorylation are required for asymmetric histone partitioning[23,26]. Consistently, live-cell imaging revealed asymmetric microtubule dynamics in dividing primary cultured HBCs. Disruption of microtubules by NZ or expression of an H3T3A mutant histone abolished asymmetries in histone inheritance, differential RNA Pol II phosphorylation, and polarized p63 association in HBCs, demonstrating conservation of key regulatory mechanisms. These findings reveal both conserved features and system-specific nuances of asymmetric histone inheritance across adult stem cell systems, from *Drosophila* to mammals, in contexts of tissue homeostasis and regeneration.

HBCs are largely quiescent under homeostatic conditions but become activated and proliferative following OE injury. During regeneration, p63 is transiently downregulated to permit its activation and subsequently upregulated in proliferating HBCs to support differentiation and self-renewal. In this study, we used methimazole (MMZ), a drug commonly used to induce OE injury and HBC activation. It should be noted that different lesion methods[11,14,15] may activate HBCs with different temporal dynamics and even potentially distinct mechanisms. Based on our results with methimazole (Fig. 1B, C, Supplementary Fig. 1), activated HBCs likely undergo only one to two divisions during regeneration. The resulting daughter cells can differentiate into globose basal cells or sustentacular cells[12], while also replenishing the HBC pool for future repair. How individual HBC divisions coordinate to accommodate these two functions remains to be explored in the future.

In many adult tissues, stem cells divide asymmetrically to balance self-renewal and differentiation[58]. In injured OE, we found that ~40% of activated HBCs divide asymmetrically with respect to histone inheritance and p63 distribution. Notably, sister chromatids with higher histone levels also exhibited increased p63 abundance, whereas sister chromatids with lower histone levels displayed reduced p63 levels. These findings suggest that asymmetric histone inheritance may act as a chromatin-based guide to differentially prime daughter cell fates by regulating both the chromatin association of key fate determinants, such as p63, and transcription reinitiation. Indeed, asymmetric histone distribution correlated with differential enrichment of Pol IIS5ph and Pol IIS2ph, as well as EU-labeled nascent transcripts (Fig. 3D–F).

During late telophase, when chromosomes start to decondense, guider histones, including H4, H3, and H3.3, were asymmetrically distributed between sister chromatids (Fig. 2C–E). The sister chromatid inheriting higher guider histones and greater p63 initiated

transcription earlier, potentially enabling the corresponding daughter cell to reach a differentiation-competent state sooner. The other sister chromatid, inheriting lower guider histones and less p63 initiated transcription later, which may delay its future daughter cell to reach this transition stage (Fig. 6G). After division, the daughter cell inheriting more guider histones will start differentiation toward a progenitor or a sustentacular cell sooner for prioritizing regeneration needs, consistent with the observation of high histone levels in differentiating progenitors (p63-negative cells, Fig. 1E). The other daughter cell inheriting less guider histones will serve to replenish the stem cell pool or to differentiate depending on the sufficiency of the earlier-generated progenitors for tissue repair. The scRNA-seq results are consistent with the output of differential cell fate priming due to asymmetric histone inheritance, where the daughter carrying higher H3.3 levels preferentially expressed the differentiation-related genes, while the other daughter carrying lower H3.3 levels showed no preference for self-renewal or differentiation (Fig. 4H). It should be noted that although high p63 levels are required to maintain HBC quiescence, quiescent HBCs retain relatively low levels of histone H4 (Fig. 1E). This suggests that p63 may regulate distinct cellular states through context-dependent chromatin statuses. How p63 functions in mitotic HBCs, when high (H3.3-H4)$_2$ guides p63 enrichment, and how p63 maintains quiescent HBCs during interphase, when low (H3-H4)$_2$ associates with p63 abundance, remains unclear and may involve temporally and cell-cycle-dependent regulation of chromatin architecture.

In *Drosophila* male germline, disruption of asymmetric histone inheritance leads to defects such as tumors or stem cell hyperplasia, stem cell loss, and reduced fertility[26,27]. Similarly, in the mammalian olfactory system, disruption of asymmetric histone inheritance through NZ treatment or H3T3A mutation impairs the balance between HBC differentiation and self-renewal. This imbalance further results in attenuated OE regeneration and impaired smell recovery. Previous studies showed that Notch signaling and Rac1 could regulate cell fate decisions after injury in this system[59,60]. Future studies will be needed to elucidate the relationships between histone asymmetry and these signaling pathways in regulating HBC fates.

In summary, our results demonstrate that asymmetric histone inheritance regulates transcription reinitiation and guides distinct cell fates during mouse olfactory tissue regeneration. These studies provide insights into how epigenetic mechanisms shape stem cell behavior in vivo and highlight its importance in neural tissue repair. Understanding these processes may have broader implications for olfactory dysfunction associated with injury, aging, neurodegenerative disease, and viral infections such as COVID-19.

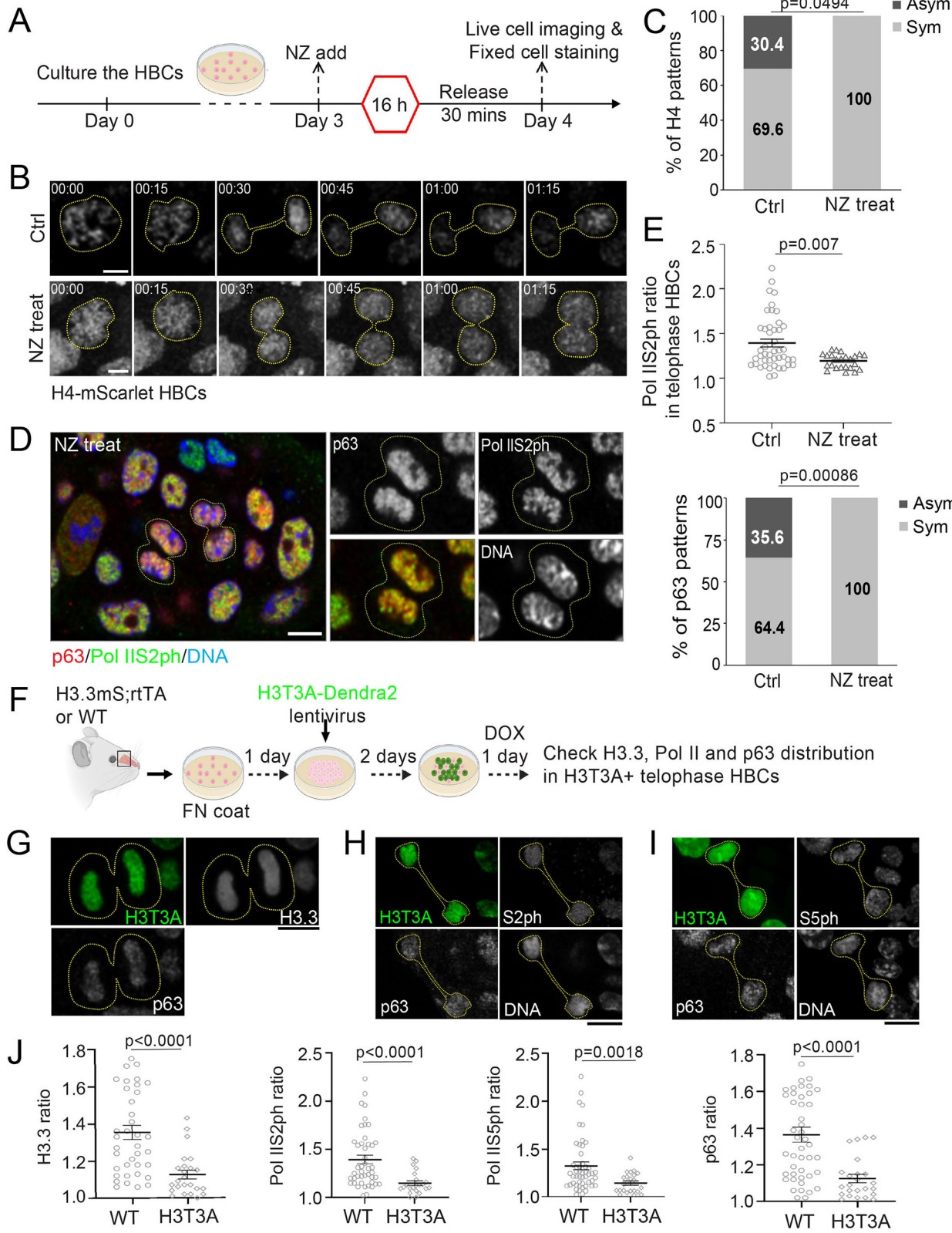

## Methods

### Mouse lines

All experiments were performed on adult mice at 6-8 weeks of age. Both males and females were used in all experiments. The mice were maintained with open access to food and water under a 12-hour light/12-hour dark cycle. TRE3G-H4-mScarlet and TRE3G-H3.3-mScarlet mouse strains were obtained from Sue Hammoud laboratory, p63-EGFP reporter mouse strain was from Sinha laboratory, and B6N.FVB(Cg)-Tg(CAG-rtTA3) 4288Slowe/J mouse strain was from Jackson lab (Strain #:016532). All animal experiments were conducted according to the procedures approved by the Johns Hopkins University IACUC guidelines (approval protocol numbers MO19A127/MO22A71/MO24A114).

**Fig. 5 | Disruption of asymmetric histone inheritance impairs asynchronous transcription re-initiation and asymmetric p63 distribution. A** Illustration of Nocodazole (NZ) treatment in primary cultured HBCs. **B** Live cell imaging showing asymmetric H4 distribution in an untreated H4-mScarlet HBC (top panel) and symmetric H4 distribution in a H4-mScarlet HBC after NZ treatment (bottom panel). **C** Percentage of H4 distribution patterns from live imaging with and without NZ treatment ($N_{Ctrl}$ = 23; $N_{NZ\ treat}$ = 10). Statistical differences according to the chi-square test. **D** Symmetrically dividing HBCs at telophase after NZ treatment. **E** Quantification for the percentage of RNA Pol IIS2ph and p63 distribution patterns in p63+ telophase HBCs. NZ treated telophase HBCs: Pol IIS2ph ratio = 1.19 ± 0.02 ($N$ = 24). Control telophase HBCs: Pol IIS2ph ratio = 1.39 ± 0.04 ($N$ = 45). **F** Illustration showing the experimental procedure of H3T3A-Dendra2 lentiviral transfection and subsequent induction of H3.3-mScalert expression in cultured HBCs. **G** Distribution pattern of H3.3-mScarlet and p63 in a H3T3A+ telophase HBC.

**H** Distribution pattern of Pol IIS2ph and p63 in a H3T3A+ telophase HBC. **I** Distribution pattern of Pol IIS5ph and p63 in a H3T3A+ telophase HBC. **J** Quantification of H3.3 ratio, Pol IIS2ph ratio, Pol IIS5ph ratio, and p63 ratio in H3T3A+ and WT control telophase HBCs. H3T3A telophase HBCs: H3.3 ratio = 1.13 ± 0.12 ($N$ = 24); Pol IIS2ph ratio = 1.15 ± 0.02 ($N$ = 22); Pol IIS5ph ratio = 1.15 ± 0.02 ($N$ = 23); p63 ratio = 1.13 ± 0.12 ($N$ = 24). Control telophase HBCs: H3.3 ratio = 1.35 ± 0.23 ($N$ = 38); Pol IIS2ph ratio = 1.39 ± 0.04 ($N$ = 45); Pol IIS5ph ratio = 1.33 ± 0.04 ($N$ = 50); p63 ratio = 1.37 ± 0.04 ($N$ = 45). All ratios: Mean ± SEM. Statistical differences according to Chi-square test in (**C**) and (**E**-bottom panel). Statistical differences according to two-sided Mann-Whitney test in (**E**-top panel) and (**J**). Scale bars: 5 μm in (**B**) and (**D**); 10 μm in (**G**)–(**I**). The cartoon elements are created in BioRender. Chen, X. (2026) https://BioRender.com/4e6v2wg. Source data are provided as a Source Data file.

## Cryosection of injured OE
Mice were euthanized by carbon dioxide ($CO_2$) asphyxiation, followed by cervical dislocation to ensure death prior to tissue collection on designated days following methimazole (50 mg/kg) i.p. injection. Half heads, with an intact olfactory epithelium, were collected for cryosection. The samples were first rinsed in PBS and fixed with 4% PFA overnight. The samples were then treated with 0.5 M EDTA for 1 day for decalcification, followed by 10% sucrose overnight and then 30% sucrose overnight for cryoprotection. All sample fixation and treatment procedures were done at 4 °C. The sample was then embedded in O.C.T and stored at −20 °C, and cut into 8–10 μm slices using a cryostat and collected onto microscope slides (12-550-15, Fisher Scientific).

## Immunofluorescence staining
For tissue sections, the slides were first boiled in 0.01 M citrate buffer (pH 6.0) for 20 minutes for antigen retrieval, followed by treatment with 0.1 M glycine in phosphate-buffered saline (PBS, pH 7.6) for 10 minutes. The OE sections were then treated with PBS containing 0.5% Triton X-100/0.05% Tween-20 (PBST) for 30 minutes and blocked in 5% BSA in PBST (blocking buffer) for 1 hour. The sections were then incubated with primary antibodies diluted in blocking buffer at 4 °C overnight, followed by detection with Alexa secondary antibodies (Invitrogen) as described[28].

For cultured cells, the samples were fixed in 4% PFA for 30 minutes. The cells were then treated with PBS containing 0.5% Triton X-100/0.05% Tween-20 (PBST) for 30 minutes and blocked in 5% BSA in PBST (blocking buffer) for 1 hour.

The primary antibodies and dilutions used were as follows: mouse monoclonal anti-p63, 1:100 (D-9, sc-25268; Santa Cruz); rabbit monoclonal anti-Ki67, 1:250 (NB600-1252, NOVUS); rabbit polyclonal anti-RNA polymerase II RPB1 phospho S2, 1:800 (AB5095, Abcam); rabbit polyclonal anti-RNA polymerase II RPB1 phospho S5, 1:800 (AB5131, Abcam); rat monoclonal anti-RNA polymerase II CTD, 1:200 (61082, Active Motif); rabbit polyclonal anti-Histone H4, 1:200 (Ab10158, Abcam), rabbit monoclonal anti-Histone H3, 1:200 (A17562, ABclonal); rabbit monoclonal anti-Histone H3.3, 1:250 (NBP2-67530, NOVUS); ChromoTek Histone-Label Atto488 (for H2A-H2B), 1:400 (tba488, Proteintech); rabbit monoclonal anti-phospho-Histone H3Thr3 (JY325), 1:800 (05-746 R, Millipore Sigma); Goat anti-Olfactory Marker Protein (OMP), 1:500 (544-10001, WAKO). Nuclei were counterstained using Hoechst 33342, and slides were mounted with VectaShield mounting medium (Vector Laboratories, Inc.).

## Western blot analysis
Proteins were extracted from one side of OE tissue using 150 μL of RIPA lysis buffer containing Protease Inhibitor (P8340, Sigma). The tissue sample was physically disrupted using electronical homogenizer and then treated with ultrasonication in the extraction buffer before centrifugation at -14,000 g at 4 °C for 10 minutes. The supernatant was collected and boiled in 1×SDS loading buffer for 5 minutes. Western blot analysis was performed as previously described[28]. The proteins were separated by NuPAGE™ Bis-Tris Mini Protein Gels, 4–12%, 1.0–1.5 mm (NP0321BOX, Thermo Fisher), and then transferred to nitrocellulose membranes (Life Sciences). After blocking, the membranes were incubated with anti-H4 antibodies (1:10,000, Sigma, 05-858) at 4 °C overnight, and after washing, then incubated at room temperature with horseradish peroxidase-conjugated (HRP)-conjugated secondary antibodies (1:1000, CST, 7074S) at room temperature for 1 h. The membranes were developed using SuperSignal™ West Pico PLUS Chemiluminescent Substrate kit (Cat# 34577, Thermo Fisher). The immunoreactive protein bands were visualized using a G:Box Chemi XRQ gel doc system (Syngene).

## EdU pulse-labeling of mouse OE tissue
EdU labeling was performed using the Click-iT Plus EdU Imaging Kit (Invitrogen C10640 and C10339) according to the manufacturer's instructions. 6-8-week-old mice were injected i.p. with 100 μL of 2.5 mg/mL EdU dissolved in PBS. The OE tissues were collected 2 hours after EdU injection. The samples were fixed and cryo-sectioned as described in the cryosection preparation, followed by immunostaining. Fluorophore (Alexa Fluor 647 or 594) conjugation to EdU was performed prior to secondary antibody incubation.

## Primary HBC culture with extracellular matrix
The primary HBC culture was performed based on the previously published method[39] with optimized culture conditions. Briefly, OE tissues were dissected from adult mice (6-8 weeks old) 2 days after the methimazole injection. The collected OE tissue was finely minced in the complete HBC culture medium and then treated with 1 × collagenase/hyaluronidase at 37 °C for 45 minutes. Samples were centrifuged at 1200 g for 5 minutes, and the cell pellets were resuspended in 5 mL pre-warmed 0.25% TrypLE (12604013, Thermo Fisher) at 37 °C for 2 minutes followed by the addition of 10 mL of ice-cold PBS. The samples were then centrifuged at 1200 g for 5 minutes, resuspended in 5 mL of pre-warmed dispase (07923, Stemcell Tech) and 10 μL of DNase 1 (07900, Stemcell Tech), and briefly triturated at 37 °C for 1 minute. Then, 10 mL of ice-cold PBS was added, and samples were filtered through Falcon 40 μm cell strainers into 50 mL conical centrifuge tubes. The cells were then collected and cultured in the complete HBC medium. To maintain HBCs in different activation statuses, two types of extracellular matrix, Tropoelastin (07003, Stemcell Tech) and Fibronectin (F2006, Sigma), were used. Before seeding the cells, the culture dishes were coated with Tropoelastin or Fibronectin overnight.

## Live cell imaging of HBCs
The cultured HBCs were infected with lentivirus (Lenti-CAG-FUCCI-CA5, from HHMI core facility) at -10 MOI in the HBC medium for 1 day and then maintained in fresh HBC medium for 1 more day to allow for fluorescent reporter expression. Live cell imaging of HBCs was

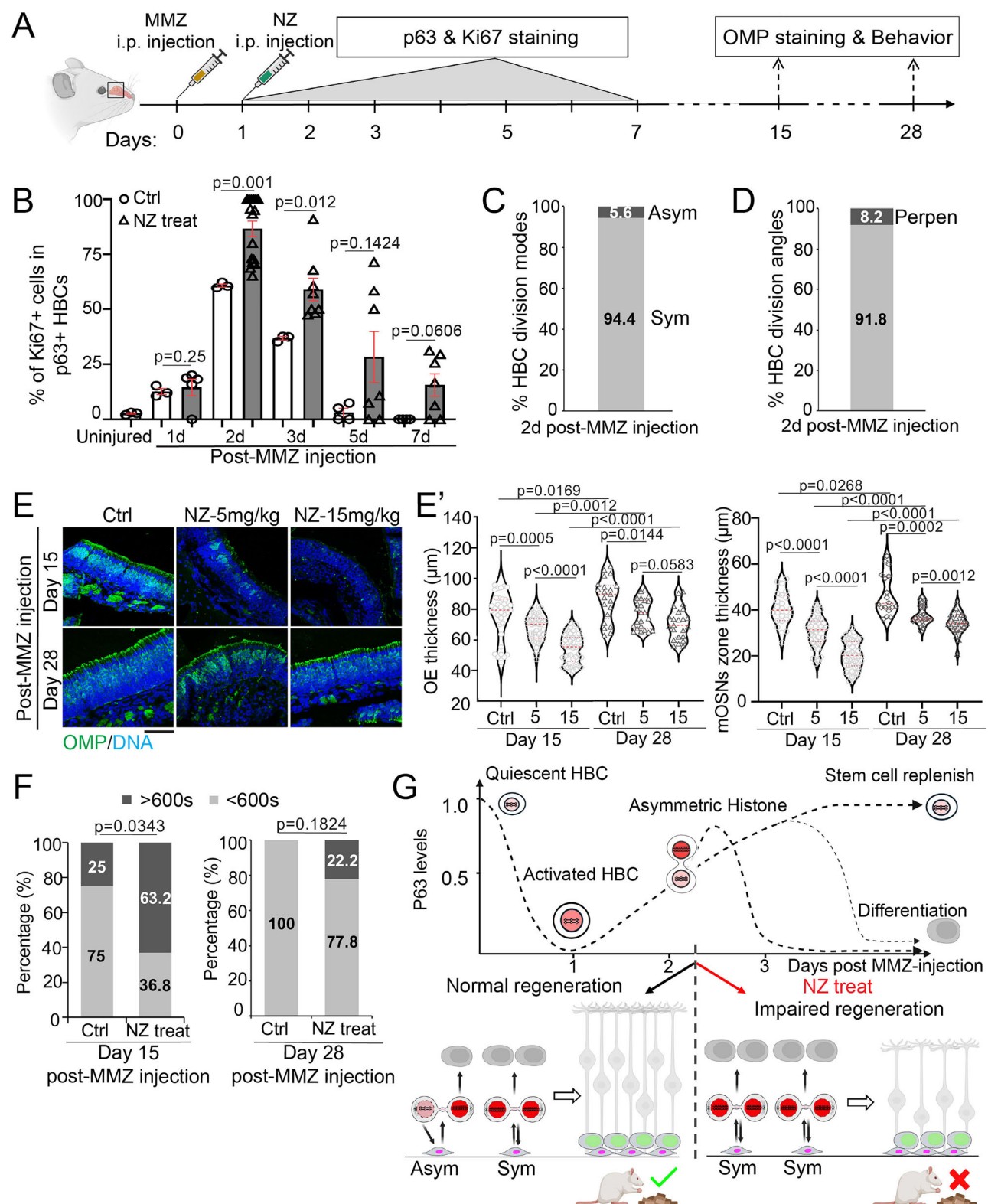

performed on a STELLARIS 5 confocal microscope equipped with a Leica HC PL APO 63× 1.40-NA oil CS2 objective, a white light laser, and a power HyD S detector. The objective was maintained at 37 °C throughout the experiment using an objective heater. 5% $CO_2$ and 100% humidity were maintained by the Tokai Hit STX stage top incubation system. The same imaging procedure was followed for imaging of the live tubulin (SPY555-tubulin) and live DNA (SPY555-DNA) dye.

## H3T3A mutation in primary culture HBCs

The H3T3A-Dendra2 construct was generated based on the previously described H3-Dendra2 plasmid[28]. Briefly, the threonine-to-alanine point mutation at residue 3 of histone H3 (H3T3A) was introduced by site-directed mutagenesis (NEB #E0554). The resulting H3T3A-Dendra2 sequence was verified by Sanger sequencing.

**Fig. 6 | Disruption of asymmetric histone inheritance impairs OE regeneration and attenuates olfactory behavior recovery. A** Illustration of experimental design for the disruption of asymmetric histone inheritance during the early OE regeneration after injury with Methimazole (MMZ) injection at Day 0 and Noco-dazole (NZ) injection at Day 1 in adult mice. p63 and Ki67 staining are performed in the early regeneration stage (Day 1–7). OMP staining and food-seeking behavior are performed in the middle and late regeneration stages (Day 15 and Day 28). **B** Comparison of percentage of proliferating (Ki67+) p63+ HBCs from OE on Day 2 post-MMZ injection with or without NZ injection (Fig. 1B, Supplementary Fig. 1). NZ dose was 15 mg/kg. Statistical differences according to the two-sided Mann-Whit-ney test. $n$ = 3 biological replicates and error bars represent mean ± SEM. **C** Ratios of asymmetric *versus* symmetric p63 distribution in p63+ telophase HBCs ($N$ = 36) from NZ-treated OE. **D** Quantifications of division angles in p63+ anaphase/telo-phase HBCs ($N$ = 49) from NZ-treated OE. **E** OE sections stained with OMP (mOSN marker) on Day 15 (top panel) and Day 28 (bottom panel) post-MMZ injection with NZ dose of 5 mg/kg and 15 mg/kg, respectively. Scale bars: 50 μm. **E'** Quantification

for the thickness of the whole OE (left panel) and OMP+ mOSN zone (right panel), respectively. Statistical differences according to the two-sided Mann-Whitney test. **F** Food seeking behavioral results on Day 15 ($N_{Ctrl}$ = 16; $N_{NZ\ treat}$ = 19) and Day 28 ($N_{Ctrl}$ = 7; $N_{NZ\ treat}$ = 9) post-MMZ injection. The NZ dose was 15 mg/kg. Statistical differences according to the chi-square test. **G** Cartoons depicting the histone-guided asymmetric cell fate specification and its biological consequence during OE regeneration (see Discussion). Bottom panel: (Left) Normal regeneration, where activated HBCs divide with either symmetric (60–70%) or asymmetric (30–40%) histone inheritance to generate progenitors or replenish the HBC pool, resulting in a balanced cell fate of the HBC daughters, which allows OE and olfactory behavior to recover in a timely manner. (Right) Regeneration with NZ treatment, where activated HBCs divide with only symmetric histone inheritance, resulting in imbalanced cell fates of the HBC daughters, which impairs OE regeneration and attenuates olfactory behavior recovery. HBCs are indicated in magenta, and GBCs are in green. The cartoon elements are created in BioRender. Chen, X. (2026) https://BioRender.com/nvwdduj. Source data are provided as a Source Data file.

---

The H3T3A-Dendra2 lentivirus was generated by the Janelia Viral Tools team (RRID:SCR_026440). Briefly, the H3T3A-Dendra2 coding sequence was subcloned into a lentiviral expression vector under the control of CAG promoter (ALSTEM, Inc., Cat LV300). Lentivirus was produced by co-transfecting the H3T3A-Dendra2 lentiviral plasmid together with packaging plasmids (Rev, Gag/Pol, and VSV-G) into HEK293T cells using a standard polyethylenimine transfection method. Viral supernatants were collected at 48 and 72 hours post-transfection, purified by ultracentrifugation through a 20% sucrose cushion, and filtered through a 0.22 μm membrane. Viral titer was measured by quantitative PCR and determined to be $2.9 \times 10^9$ IU/mL. Aliquots (10 μL) were stored at −80 °C until use.

For transfection, primary cultured HBC cells in one FluoroDish with 35 mm Diameter and 23 mm well (FD35-100, WPI) were incubated with 500 uL culture medium with 10 uL H3T3A-Dendra2 lentiviral particles, followed by fresh medium replacement after 24 hours. Expression of H3T3A-Dendra2 was visualized 48 hours post the transfection.

### EU pulse-labeling in primary culture HBCs
Primary cultured HBCs were labelled with 0.5 mM 5-ethynyluridine (EU) in complete HBC medium for 1 hour at 37 °C. Then the media with EU was removed, and the cells were washed 3 times with PBS and fixed with 4% PFA overnight at 4 °C. To visualize EU, HBCs with primary antibodies staining were subsequently processed for the click reaction with Alexa A555-azide (ThermoFisher, A20012) and then incubated with secondary antibodies.

### Imaging acquisition
Samples were imaged under a Leica SPE confocal microscope or STELLARIS 5 confocal microscope equipped with a Leica HC PL APO 63× 1.40-NA oil CS2 objective, a white light laser, and a power HyD S detector. Z-stacks of 0.5 μm per layer were taken for mitotic HBCs. The ImageJ software was used to quantify the fluorescent intensities and evaluate the division angle of anaphase and telophase HBCs.

### 3D quantification of telophase HBCs
To quantify the total amount of proteins (such as histones, p63, and RNA Pol II), we conducted a 3D quantification in volume by measuring the fluorescent signal in each plane from the Z-stack[30,40]. Specifically, the 3D quantification of the fluorescent signal was done manually using ImageJ. Un-deconvolved raw images as 2D Z-stacks were saved as unscaled 16-bit TIF images, and the sum of the gray values of pixels in the image ("RawIntDen") was determined using ImageJ. The gray values of the fluorescent signal pixels for each Z-stack were calculated by subtracting the gray values of the background signal pixels from the

gray values of the raw signal pixels. The total amount of the fluorescent signal in the nuclei was calculated by adding the gray values of the fluorescent signal from all Z-stacks. The total amount of fluorescent signals would represent the total amount of protein across the entire nucleus and be used for quantifying the asymmetric and symmetric protein distribution in telophase HBCs.

### Division angle measurement
To measure a division angle of HBCs, use the ImageJ - "Angle" tool to click on three points that define the angle you want to measure. As shown in Supplementary Fig. 3A, the olfactory basal membrane was used to define the baseline, and the division axis was estab-lished by the two sets of sister chromatids. During analysis, the division angles of anaphase and telophase HBCs at 60°–90° to the basal membrane were classed as perpendicular; those that were at 0°–30° were classed as parallel. The perpendicular and parallel divisions were used for the quantification in Supplementary Fig. 3C.

### Co-localization analysis
The co-localization assay was performed using FIJI (ImageJ) software. The image was imported into ImageJ, and the HBCs were identified via P63 fluorescent signals. Drawing near the edge of the cell to the best of our ability, we outlined the HBCs of interest using the "freehand selections" tool on the toolbar. After outlining the cell, the selected area was duplicated using a function under the toolbar (Image>Du-plicate Image). The duplicated area is then split into individual chan-nels using the toolbar (Image>Color>Split Channels). With this, the images are ready for the co-localization analysis using the Coloc 2 plugin (Analysis>Co-localization>Coloc 2). The Coloc 2 tool imple-ments and performs the pixel intensity correlation over space (pixel intensity spatial correlation analysis). After opening the Coloc 2 plugin pop-up, select the two desired channels to perform the analysis on into "Channel 1" and "Channel 2." We used Spearman's Rank Correlation value to compare co-localization between different datasets. The result is +1 for perfect correlation, 0 for no correlation, and −1 for perfect anti-correlation. The co-localization analysis was performed between p63 and different histones (H4, H2A-2B, H3.3, and H3) in Fig. 2H.

### Single-cell RNA-seq and data analysis
The single-cell RNA-seq (scRNA-seq) experiment was done following the G&T-seq protocol[47]. Single daughter cells after division of ex vivo cultured primary HBC cells were collected individually using a mouth pipette with pairing information noted. 1 mL of pre-diluted ($1:10^6$) ERCC spike-in (Invitrogen Cat# 4456740) was added to the single cell lysate. RNA in the lysate was captured by oligo-dT beads and subjected to reverse transcription. cDNA from a single cell

was amplified by 18 PCR cycles before dual indexing with the Nextera XT kit (Illumina Cat# FC-131-1096). The quality of single-cell libraries was confirmed by TapeStation, and libraries of 96 cells from 48 HBC pairs were pooled together for sequencing (150 bp paired-end) in one lane on the Illumina NovaSeq X Plus platform (Novogene US).

The quality of all FASTQ files from Illumina sequencing was analyzed and confirmed by FastQC (v0.12.1). scRNA-seq reads were trimmed with Trimmomatic (v0.39)[61]. The trimmed RNA-seq reads with both pair mates were aligned by STAR (v2.7.11a)[62] to the GRCm38 annotation (ENSEMBL release 100) plus ERCC information. StringTie (v2.2.1)[63] was used to generate counts of genes in the GTF reference. After aligning all the RNA-seq reads of the 96 (48 pairs) single cells to the reference genome, the total number of different genes detected in every single cell was examined, and only the cells that captured more than 10,000 genes would be kept for downstream analysis. Meanwhile, the cells that have more than 5% of the reads aligned to mitochondrial genes would be eliminated from downstream analysis. All 96 cells met the quality control criteria and were retained for downstream analysis.

Publicly available single-cell RNA-seq datasets of in vivo retrieved HBC cells were downloaded from Gene Expression Omnibus (accession numbers: GSE99251, GSE95601)[12,46]. scRNA-seq count matrices of all single cells were analyzed by Seurat (v5.1.0)[64]. The "CCAIntegration" method in Seurat was used when integrating the published scRNA-seq data into the newly generated HBC pair datasets before visualization in UMAP. The newly generated scRNA-seq dataset in this study was deposited in Gene Expression Omnibus (accession numbers: GSE286046).

After clustering of the single cells from HBC pairs by Seurat, the RNA-seq results of all cells falling into the "Activated HBC" cluster were treated as one group and compared to the data of cells from the "Renewed HBC" cluster using DESeq2 (v1.44.0)[65]. ERCC spike-in was used for normalization. Log2 fold change = ± 1 with adjusted R value < 0.1 was used as a cutoff for significantly up- or down-regulated genes. Significantly upregulated genes were then used for gene ontology analysis with DAVID[66]. ERCC normalized gene counts were used for violin plots, and R values were calculated by a two-tailed t-test.

## Mathematical modeling of the cell cycle and HBCs activation

We consider a model based on ordinary differential equations to capture OE tissue regeneration dynamics after tissue injury. At an intuitive level, the model works by considering the activation of otherwise quiescent HBCs after injury, and activated HBCs undergo asymmetric division, leading to the formation of differentiated cells. The post-injury buildup of differentiated cells reduces HBC proliferation via feedback regulation, and HBCs return to their original quiescent state upon OE regeneration, ensuring tissue homeostasis. We next describe the mathematical model in detail.

Let $H_1$, $H_2$, and $H_3$ denote the fraction of HBCs that are in the G1, S, and G2/M phases of their cell cycle at time $t$ after tissue injury. The transition from G1 to S is assumed to occur with rate $k_1$. As we will soon see, this rate $k_1$ is assumed to depend on a feedback signal from differential cells, such that the absence of differentiated cells triggers HBC activation by enhancing this rate and reducing the G1 phase duration. The transition from S to G2/M occurs with rate $k_2$ and we set this rate to be $k_2 = 1.33\ day^{-1}$, which corresponds to the duration of S phase to be $1/k_2 \approx 18$ hours consistent with single-cell microscopy data. Cells in the G2/M undergo division with rate $k_3 = 6\ day^{-1}$ corresponding to the average time spent in the G2/M phase to be 4 hours. The division of HBCs results in the formation of another HBC in G1 phase, and a differentiated cell whose population number we denote by $D$. This model results in the following system of differential equations that predict the temporal dynamics of $H_1$, $H_2$, $H_3$, and $D$:

$$\frac{dH_1}{dt} = -k_1 H_1 + k_3 H_3 \tag{1}$$

$$\frac{dH_2}{dt} = k_1 H_1 - k_2 H_2 \tag{2}$$

$$\frac{dH_3}{dt} = k_2 H_2 - k_3 H_3 \tag{3}$$

$$\frac{dD}{dt} = k_3 H_3 - \gamma D \tag{4}$$

where $\gamma$ is the turnover rate of differentiated cells. As mentioned earlier, tissue homeostasis is feedback regulated by making the HBC G1 phase duration dependent on the differential cells via the Hill equation $k_1 = \frac{k_{max}}{1 + \left(\frac{D}{K}\right)^h}$ for some positive constants $k_{max}$, $K$, and Hill coefficient $h$. The absence of differential cells $D = 0$ shortens the G1 phase duration by increasing the rate at its maximum value $k_1 = k_{max}$.

We use $H_2$, i.e., the fraction of HBCs in the S phase as the mathematical counterpart to the experimentally measured p63+/EdU+ fraction in Fig. 1C. In the absence of tissue injury, we see basal levels of HBC proliferation, with the fraction of HBCs (p63+/EdU+) seen to be roughly 6.5% in control samples. Using this as an initial condition for the model together with $D = 0$, we fit the model-predicted dynamics of $H_2$ to data by minimizing the least square error between them. We further assume an additional parameter $\tau$ that represents an initial time-delay between tissue injury and HBC activation, with $H_1$, $H_2$, and $H_3$ remaining at their basal levels for time $t < \tau$ and following the above differential equation for $t > \tau$. The fit of $H_2$ to data is illustrated in Supplementary Fig. 5D with feedback parameters estimated as $k_{max} = 2.27\ day^{-1}$, $\tau = 0.8\ day$, $h = 2$, $K = 0.4\ a.u.$ and $\gamma = 0.06\ day^{-1}$.

## Mathematical modeling of asymmetric cell division and RNA Pol II binding affinity

To model transcription re-initiation during mitotic exit, we analyze the RNA Pol II ratio between sister chromatids in HBCs during telophase. The histogram of the ratios in Supplementary Fig. 6E shows a bimodal distribution consistent with a fraction of cells undergoing symmetric cell division (i.e., equal partitioning of RNA Pol II between sister chromatids) vs. asymmetric cell division (i.e., preferential binding of RNA Pol II to one of the sister chromatids). We computed the statistics of this ratio measured across 55 single cells and obtained the mean, coefficient of variation, and skewness of the ratio to be $1.4 \pm 0.08$, $0.21 \pm 0.03$, $0.58 \pm 0.45$, respectively, where the $\pm$ denotes the 95% confidence interval as obtained by bootstrapping. We next develop a mathematical model to predict and match these statistics.

Let a cell undergo symmetric cell division with probability $p_s$ and RNA Pol II is partitioned symmetrically between the sister chromatids. In this case, we consider the RNA Pol II intensities $X_1$ and $X_2$ at the two sister chromatids to be drawn from a Gaussian distribution with a mean of one and standard deviation $\sigma$, where $\sigma$ corresponds to the technical noise in RNA Pol II intensity quantification. In this case of symmetric division, the RNA Pol II intensities and their ratio $r$ are determined as $X_1 \sim N(1, \sigma), X_2 \sim N(1, \sigma), r = \frac{Max(X_1, X_2)}{Min(X_1, X_2)}$. With probability $1 - p_s$ a cell undergoes asymmetric division, where RNA Pol II is preferentially partitioned towards one sister chromatid and the RNA Pol II intensities $X_1$ and $X_2$ are determined by $X_1 \sim N(1 - p_b, \sigma), X_2 \sim N(1 + p_b, \sigma), r = \frac{Max(X_1, X_2)}{Min(X_1, X_2)}$. where $p_b$ represents the bias in RNA Pol II preferentially binding to one sister chromatid over the other, with $p_b = 0$ corresponding to symmetric RNA Pol II partitioning. In both cases, the ratio $r$ is computed by dividing the maximum of the two random variables $X_1$ and $X_2$ by their minimum.

The proposed model has three parameters: $p_s$ the probability of symmetric cell division, $\sigma$ representing noise in RNA Pol II abundance quantification, and $p_b$ the bias in RNA Pol II binding for asymmetric cell division. For given values of $p_s, \sigma, p_b$ we simulated this model for 5,000 in-silico cells, and the statistics (mean, coefficient of variation, skewness) of the ratio $r$ obtained from simulations were matched with corresponding statistics obtained from data (as reported above). This process results in the estimation of parameters as $p_s \approx 0.45, \sigma \approx 0.09, p_b \approx 0.23$, and a sample histogram of ratios obtained from simulation is shown in Supplementary Fig. 6F. These results imply an approximately 0.55 probability of asymmetric cell division, and in this case, the ratio of RNA Pol II binding affinity to one sister chromatid over the other is given by $\frac{1+p_b}{1-p_b} \approx 1.6$, implying a 60% increase in binding affinity.

## Olfactory behavioral tests

The 6-8-week C57BL/6 J wild-type mice of both sexes were used for NZ treatment and olfactory behavioral tests.

Buried Food Seeking Behavior Test: The mouse was deprived of food for 24 hours but with an open water supply. The mouse was habituated in the test room for 30 minutes before the test and then placed into the testing cage (8 H x 10 W x 18 L, in inches) containing a piece of ~0.5 g peanut bar (Item: 1952, Costco Wholesale) buried at a random location underneath ~1.5 cm of fresh bedding. If an animal finds the peanut bar in less than 10 minutes, it is counted as a success, and the time spent finding the peanut bar is recorded. If an animal fails to find the peanut bar in 10 minutes, the test is terminated.

Olfactory Preference Test: The olfactory preference test was performed based on the published protocol[51]. Briefly, the mouse was habituated sequentially in three bedding-free clean cages (8 H × 10 W × 18 L, in inches) for 15 minutes each. Each mouse was tested for each of three odors for 3 minutes each. The mice were tested and proceeded to the next odor as a cohort. For each trial, a piece of filter paper (2 × 2 inches) containing ~0.5 mL of the odorant solution was placed at one side of the testing cage, opposite to the mouse's initial position, and then the mouse was given 3 minutes to explore the odorous filter paper. The total probe time (defined by the duration when the tip of the nose was within ~1 mm distance from the filter paper) was quantified from the recorded videos by a blind observer. The concentrations of odorant solutions are listed below: lemon extract (1:10 dilution in distilled water, VITACOST-SKU #: 051381911249); Vanilla extract (1:10 dilution in distilled water, VITACOST-SKU #: 051381912314); Peanut butter (1 g per 10 mL odorless mineral oil, Item: 555000, Costco Wholesale). All cages and materials were thoroughly cleaned with 70% ethanol and dried between trials to prevent cross-contamination.

## Statistics and reproducibility

Statistical evaluation was performed by two-sample Welch t-test, Mann-Whitney unpaired t-test, and Chi-square test. Data are presented as Average ± SEM. $P < 0.05$ was considered significant for all analyses, and the exact $p$-values were directly indicated in the figures. Statistical analysis was performed by GraphPad Prism 10 software. The fluorescence intensity was measured by the ImageJ software. At least three mice were used as the biological replicates for the experiments, including tissue cryosection and staining. For the olfactory behavioral tests, 16 control and 19 NZ-treated mice were used for the buried food seeking behavior test on Day 15 post-MMZ injection, and 7 control and 9 NZ-treated mice were used for the buried food seeking behavior test on Day 28 post-MMZ injection. 8 control and 10 NZ-treated mice were used for the olfactory preference behavior test on Day 15 and Day 28 post-MMZ injection. No data was excluded except the serial dilution results of H4-mScarlet Western Blot. As clarified in the Fig. 1F legend, the first two lanes were used for quantification due to the low signals of 1:4 dilution lane. All experiments were replicated at least three times independently to assure reproducibility. For each experiment, both control and treated groups were randomly selected. Sample preparation, data collection, and analysis were performed blindly.

## Reporting summary

Further information on research design is available in the Nature Portfolio Reporting Summary linked to this article.

## Data availability

The scRNA-seq dataset in this study was deposited to Gene Expression Omnibus under the accession code GSE286046. Source data are provided with this paper. Additional materials and protocols supporting this study are available from the corresponding authors upon reasonable request. Source data are provided with this paper.

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

## Acknowledgements

We thank the Howard Hughes Medical Institute at Janelia Research Campus Viral Tools Shared Resource Core Facility (RRID:SCR_026440) for the virus support, Drs. Shukry Habib, Nicolas Plachta, Yanxiang Deng, and the Chen and Zhao laboratory members for their critical comments and suggestions on this work. This work has been supported by NIH F31HD100124 (G.M.), NIH 1DP2HD091949-01 (S.S.H.), Open Philanthropy Grant 2019–199327 (5384) (S.S.H.), NIH R01AR073226 (S.S.), NIH R35GM148351 (A.S.), NIH R01DC016065 (H.Z.), Howard Hughes Medical Institute (X.C.) and Gorden and Betty Moore Foundation (B.M.).

## Author contributions

Conceptualization: B.M., H.Z., and X.C. Methodology: B.M., G.Y., and A.S. Investigation: B.M., G.Y., J.Y., C.W., and J.P.V. Data analysis: B.M. and G.Y. Resources: G.M., S.S.H., and S.S. Manuscript writing: B.M., G.Y, H.Z., and X.C. Supervision: H.Z. and X.C. Funding acquisition: G.M., S.S.H., S.S., B.M., H.Z., and X.C.

## Competing interests

The authors declare no competing interests.
