## [Transparent Peer Review file · Nature Communications]

Asymmetric Histone Inheritance Regulates Olfactory Stem Cell Fates During Regeneration

Corresponding Author: Dr Xin Chen

Version 0:

Reviewer comments:

Reviewer #1

(Remarks to the Author)

The paper by Ma et al. is a comprehensive assessment of a very basic biological question – asymmetry of outcome by daughters of stem cells – in a particularly tractable mammalian stem cell and tissue, namely the HBCs of the olfactory epithelium (OE). The HBCs being subject to regulation by the TF p63 evince a dynamic relationship to OE status. They take advantage of in vitro and in vivo approaches and use a primarily correlative protein distribution approach to p63, the various histones, and the machinery for transcriptional initiation. The data seem solid with some exceptions. One concern is the use of fluorescently tagged H4 in a transgenic mouse line given the absence of a decent antibody for mapping H4. Can the authors provide assurance that the tag does not materially affect H4 biology? The added molecular weight associated with the tag accentuates that concern. The demonstration of symmetric vs asymmetric division modes and the correlation with histone distribution is striking.

Are the cells in culture that diminish p63 actually doing anything? Differentiating? Still plastic? What evidence can the authors provide that this has to do with activation of HBCs as opposed to re-establishment? Some form of functional assessment, such as differentiative capacity after transplantation would strengthen the association between the changes in p63 and functional outcome.

The in vitro and in vivo experiments with nocodazole disruption of microtubule-based distribution are exciting, but the authors don't offer a way to assess off-target effects on the cells. That consideration of off-target effects is needed to conclude definitively that the disruption of asymmetry is the cause of the substantial differences in OE regeneration and olfactory-guided behavior in vivo. Other types of HBC manipulation – of Notch signaling and of Rac distribution after injury – also cause an alteration in the balance of types of HBC progeny (Louie et al, 2023; 2024). It is exciting to ponder whether histone asymmetry is the ultimate means by which these diverse manipulations have their effect, but these alternatives need to be considered and controlled for. In addition, food-finding is a relatively crude behavioral measure of olfactory function. (What is a peanut bar, BTW? Perhaps provide a product name.) One wishes the authors had employed a second test to validate the result.

There are some deficiencies in the paper. The authors do a poor job of describing the state of the field in the introduction to the paper. The olfactory epithelium contains TWO stem cell populations: the normally dormant HBCs and a subset within the heterogeneous population of GBCs, which contribute to the continuously active generation of neurons under homeostatic and even accelerated neurogenesis. The latter has been established through functional assessments of tissue stemness and tentatively identified on the basis of multiple molecular markers. So that the readers who are not expert in olfactory biology do NOT come away with the misapprehension that HBCs are THE stem cell of the OE (which error, unfortunately, has not been eliminated from the literature).

In addition, there is a fundamental complexity to the response to lesion here that the authors do not address. At day 1 after methimazole injury p63 levels are essentially nil, which comports with previous observations using methyl bromide lesion (in which the intoxication is more time-locked and limited). As a consequence, all the HBCs should begin to activate from their dormancy. By looking at day 2 and focusing on cells that have some level of p63, the events they are studying are slightly different than pure activation and indicative of how context may be influencing the downstream events of activation. This is a subtle difference but may be highly relevant and/or specific to the form of lesion. Again, to ensure that the field comes away

with a true understanding of how the regenerative process unfolds, these complexities need to be presented by the authors.

There are a few minor problems with aligning text, figure legends, and figures. 1) First citation for Fig 1E and 1F in the text are inverted with respect to the labeling of the figure panels. 2) Figure legend S1 is wrong and does not correspond correctly with the panels in the Figure. 3) Please add labels for the markers in the combined image in Figure 3. 4) Also in Figure 3 – some cells connected by dotted lines are clearly in DIFFERENT clusters, despite what is said in the Figure Legend.

Reviewer #2

(Remarks to the Author)

This study reveals that asymmetric histone inheritance regulates cell fate specification in mammalian adult stem cells during tissue regeneration. Using the mouse olfactory epithelium (OE) model, the authors demonstrate that histones H4, H3, and variant H3.3—but not H2A-H2B—are asymmetrically distributed in 30–40% of dividing horizontal basal cells (HBCs) during injury-induced regeneration. This asymmetry correlates with unequal partitioning of the transcription factor p63 and asynchronous transcription re-initiation (evidenced by asymmetric RNA Pol IIS2ph/S5ph), priming daughter cells for distinct fates. Functional disruption via nocodazole (NZ) abolishes histone/p63 asymmetry, impairs OE regeneration, and delays olfactory recovery, underscoring the essential role of epigenetic inheritance in tissue repair. Single-cell RNA sequencing of paired HBC daughters further confirms asymmetric multilineage priming, linking H3.3 transcript asymmetry to divergent differentiation trajectories. These findings establish histone asymmetry as a conserved mechanism guiding stem cell plasticity in mammals, with implications for regenerative therapies targeting epigenetic regulators. But this article lacks the molecular mechanism of asymmetric histone distribution. The most critical step in histone inheritance during cell division is the redistribution of parental histones at the DNA replication fork during DNA replication.

Main comments:

1. Core Issue: Lack of Molecular Mechanism for Asymmetric Histone Segregation

The study demonstrates asymmetric histone (H4/H3/H3.3) inheritance in HBCs but fails to address how histones are directionally segregated during DNA replication—the most critical step for epigenetic inheritance. In mammalian cells: DNA replication initiates at multiple origins, with parental histones redistributed to both leading and lagging strands at each fork (Xu et al., Science 2010). Symmetry challenge: This bidirectional redistribution should theoretically equalize parental histone distribution between sister chromatids, making the observed asymmetry (30–40% of divisions) mechanistically puzzling. The authors neither characterize replication-coupled histone recycling nor provide evidence for biased histone transfer (e.g., strand-specific retention or chaperone-mediated sorting), leaving a gap between the observed asymmetry and established replication biology.

2. The claimed asymmetry conflicts with prior work showing: equal histone partitioning: Canonical histones (H3-H4)₂ are randomly deposited on both daughter strands during replication (Petryk et al., Science 2018). (H3.3-H4)₂ are symmetrically deposited on both daughter strands during replication, as well (Xu et al., Nature communications 2022). H3.3 dynamics: While H3.3 is replication-independent, its asymmetric inheritance (Fig. 2E) lacks mechanistic explanation. The study does not distinguish whether asymmetry arises from de novo deposition (post-replication) or replication-coupled recycling.

3. To reconcile their findings with replication biology, the authors should: Track histone dynamics during S-phase: Use pulse-chase labeling (e.g., Click-IT EdU combined with H3-H4 immunoprecipitation) to map parental histone redistribution at replication forks in HBCs. Compare histone retention on leading vs. lagging strands via eSPAN. Validate in replication mutants: Perturb replication symmetry (e.g., MCM2 mutants that bias histone retention) and assess asymmetry persistence.

4. If replication-independent mechanisms drive asymmetry, the authors must: Rule out technical artifacts: Overexpressed H4-mScarlet/H3.3-mScarlet (Fig. 1D, 2E) may disrupt native chromatin recycling. Endogenous tagging (e.g., C-terminal tags) is needed. Clarify temporal control: Asymmetry might emerge post-replication, but this is not addressed.

5. related to 4. Overexpressed H4-mScarlet cannot reflect the exact distribution of endogenous H4. The chromatin of the cell who acquires more H3.3-H4 tends to transcribe and be looser, then the density of nucleosomes should be lower. How could the cell acquire more H4? The explanation of these results could not make sense.

6. Technologically, how did the author ensure that the two daughter cells in the tissues are in the same horizontal plane while preparing the slices? If the focus of the two daughter cells is not in the same plane when taking pictures, the one out of focus looks like with a weak signal, resulting in the observed asymmetry signal of detected proteins. But this is not the fact; it is only a systematic error.

7. Figure 5B, is the division of cells affected by NZ treat? NZ is usually used as a G2 cell cycle blocker. The phenotype should rule out the cell cycle arrest caused by NZ.

Reviewer #3

(Remarks to the Author)

I co-reviewed this manuscript with one of the reviewers who provided the listed reports. This is part of the Nature Communications initiative to facilitate training in peer review and to provide appropriate recognition for Early Career

Researchers who co-review manuscripts.

Reviewer #4

(Remarks to the Author)

This manuscript investigates asymmetric histone inheritance in the context of adult mammalian stem cell fate specification. The authors identify that in mouse olfactory epithelial horizontal basal cells (HBCs), canonical histones H3 and H4, as well as the variant H3.3, are inherited asymmetrically during regeneration. This asymmetry is associated with differential p63 expression and asynchronous transcriptional re-initiation between daughter cells. Functional perturbation using nocodazole shows that disrupting histone asymmetry impairs olfactory epithelium (OE) regeneration. These findings suggest a novel role of histone inheritance in tissue regeneration. There are some issues regarding clarity and quality controls that need to be addressed.

Major concerns:

The citations of Figures 1E and 1F are wrong.

The conclusion that “the fusion protein accounts for approximately 10% of endogenous H4” from the Western blotting of Figure 1E is not convincing. Quantification of the WB from three repeats and a serial dilution of the samples to compare the amount of proteins is needed.

Figure S2B-C, what does the “control” mean? Please specify the condition. In addition, an explanation of the mScarlet signal in controls needs to be addressed.

Figure 1G and others, please add the line to the cells to indicate how the intensity is measured.

The Pol II S2p and S5p are identified to be asymmetrically distributed. How about the total amount of Pol II?

The description of Figure 4B doesn't match the Figure.

For the scRNA-seq, quality controls need to be presented, and how to filter the low-quality cells. How many batches of this scRNA-seq are performed, as the scRNA-seq needs biological replicates?

In the scRNA-seq, does the expression of p63 follow the asymmetric distribution as detected in other figures?

Figure 4H, “The other daughter cell carrying lower H3.3 levels show tendency either for self-renewal (higher p63, Trp53inp1, Myc) or for differentiation (lower Krt5, Krt14, Icam1) (Fig. 4H).” I'm confused that in previous results, H3.3 is high in P63 high cells, but why is the correlation opposite in scRNA-seq?

Figure 4I, from the PCA plot, these renewed and activated HBC do not separate into two clusters as indicated in the results section.

Figure 4L, quantification is needed to show the SUV39 staining.

Is CenH3 asymmetrically distributed in the scRNA-seq results?

The finding that H3.3 is asymmetrically inherited in contrast to *Drosophila* is intriguing. Could the authors expand on whether this is due to P63 or another mechanism that leads to this difference?

While nocodazole treatment supports a causal link, it affects microtubule dynamics broadly. Is there a way to disentangle histone partitioning from spindle orientation more specifically? Are there alternative approaches to target chromatin directly, like the authors did before, to mutate the phosphorylation sites in H3?

Minor concerns:

Figure 2B, the DIC image does not correspond to the IF image. Could separate to avoid the misleading.

The presentation of figures is not fluent, as Figure S2 is presented and then Figure S1C-E is cited.

Figure S4, The scale bars are not correct.

Figure 5A is cited at a very early stage after Figure 2.

Figure S6, P-value is missing.

Reviewer #5

(Remarks to the Author)

Version 1:

Reviewer comments:

Reviewer #1

(Remarks to the Author)

The authors have provided a HIGHLY comprehensive and compelling rebuttal to the previous reviews. For myself, I am satisfied that they have addressed all of my concerns both major and minor.

Reviewer #2

(Remarks to the Author)

The authors have addressed my questions and comments raised for the previous submission. Therefore, I support acceptance of this work.

Reviewer #3

(Remarks to the Author)

Reviewer #4

(Remarks to the Author)

The authors have competently responded to the reviewer's comments. This reviewer has no further criticisms.

Reviewer #5

(Remarks to the Author)

Point-to-point responses to Reviewers' comments

We thank the reviewers for reviewing this manuscript. We appreciate their thoughtful feedback and collective enthusiasm about this work. The reviewers' constructive comments and our responses to their concerns, detailed below, have significantly improved this study. We have comprehensively addressed the reviewers' concerns by including new experiments and results, changing text in the revised manuscript, and responding directly to the concerns in this point-to-point response.

We have revised the manuscript accordingly and marked the revised text and figure assignments in red in the revised manuscript.

Reviewer #1

The paper by Ma et al. is a comprehensive assessment of a very basic biological question – asymmetry of outcome by daughters of stem cells – in a particularly tractable mammalian stem cell and tissue, namely the HBCs of the olfactory epithelium (OE). The HBCs being subject to regulation by the TF p63 evince a dynamic relationship to OE status. They take advantage of in vitro and in vivo approaches and use a primarily correlative protein distribution approach to p63, the various histones, and the machinery for transcriptional initiation. The data seem solid with some exceptions. One concern is the use of fluorescently tagged H4 in a transgenic mouse line given the absence of a decent antibody for mapping H4. Can the authors provide assurance that the tag does not materially affect H4 biology? The added molecular weight associated with the tag accentuates that concern. The demonstration of symmetric vs asymmetric division modes and the correlation with histone distribution is striking.

We thank the reviewer for the positive comments.

We acknowledge the reviewer's concern about the use of fluorescently tagged H4 in a transgenic mouse line. Previous studies have shown that the C-terminally tagged-H4 recapitulates endogenous H4 patterns assessed by immunostaining and chromatin immunoprecipitation (Hiroshi Kimura et al. *J Cell Biol.* 2001. PMID: 11425866; Kami Ahmad et al. *Molecular Cell.* 2002. PMID: 12086617), indicating that the tag does not affect H4 localization, cell cycle kinetics and replication-dependent chromatin incorporation. In our study, the expression of the tagged-H4 was induced by Doxycycline, and we titrated the Doxycycline dose and used a low dose to achieve a detectable but low level of tagged-H4 expression. Our Western blot result showed that tagged-H4 is expressed ~6% of the endogenous H4 levels (Fig. 1F and Supplementary Fig.1B in the revised manuscript). Importantly, we compared the HBCs dynamics during OE regeneration and found no difference between H4mS (Ctrl, in which no tagged-H4 expression) and H4mS;rtTA (the tagged-H4 is expressed due to the presence of rtTA) (Supplementary Fig. 2 in the revised manuscript). Furthermore, our results of using H4 antibody staining in culture HBCs (Fig. 2C, C' in the revised manuscript) showed similar pattern and percentage of asymmetric H4 distribution to that in telophase HBCs

expressing tagged H4 *in vivo*. Thus, we conclude that the induced expression of tagged-H4 does not affect the H4 biology in our system. Notably, a key advantage of using fluorescently tagged H4 rather than antibodies in immunostaining is that it eliminates concerns about antigen accessibility, which can be an issue in mitotic cells when chromosomes are highly condensed.

Are the cells in culture that diminish p63 actually doing anything? Differentiating? Still plastic? What evidence can the authors provide that this has to do with activation of HBCs as opposed to re-establishment? Some form of functional assessment, such as differentiative capacity after transplantation would strengthen the association between the changes in p63 and functional outcome.

The reduction of p63 is necessary for the HBC activation to exit the quiescence, as evidenced by the previous studies (Adam Packard et al. *J Neurosci*. 2011. PMID: 21677159; Nikolai Schnittke et al. *PNAS*. 2015. PMID: 26305958). After activation, the p63 level is upregulated, and this upregulation of p63 level may be critical to regulate the fate choice. The reviewer raised an intriguing question regarding until which stage the p63 diminished cells are still plastic and can return to the HBC fates. In this paper, however, we used p63 mainly as a marker to indicate HBC identity. We didn't track those p63 diminished cells.

Nevertheless, we attempted to address this question in culture by switching the coating condition from fibronectin to tropoelastin. HBCs were collected from mice on Day 2 post methimazole injection and cultured in fibronectin-coated or tropoelastin-coated dishes for 1, 2, 3, or 5 days. The cells on fibronectin were then re-plated to the tropoelastin-coated dish for 1.5 days before fixation and staining, as indicated in Fig. R1A. The morphology of HBCs changed drastically from fibronectin-coated dishes to tropoelastin-coated dishes (Fig. R1B), which is consistent with our previous data showing more domed morphology of HBCs on Tropoelastin, while flatten morphology on fibronectin (Fig. 2A in the revised manuscript). We found that HBCs in fibronectin-coated dishes exhibited significantly lower p63 levels than HBCs in tropoelastin-coated dishes (Fig. R1C). The observation that HBCs display higher p63 levels after re-plating to tropoelastin-coated dishes indicates that HBCs with diminished p63 can regain high p63 levels (Fig. R1C, D). This plasticity of p63-diminished HBCs can be maintained for 3 days in fibronectin-coated dishes. We further quantified the percentage of p63+ cells in the colonies and found that the percentage of p63+ cells drastically declined from 73.0% of 3 days to 27.2% of 5 days on fibronectin (Fig. R1E), suggesting that the HBCs cultured on FN coating condition for 5 days significantly lost the plasticity. We decided not to include this piece of data in the revised manuscript, given its preliminary and out-of-scope nature. We haven't investigated the differentiation capability of p63-diminished HBCs.

In this study, we proposed a model (Fig. 6G in the revised manuscript) that includes p63 as a fate indicator and a potential fate determinant accompanied by histone inheritance information. In the model, the sister chromatid with higher guider histones and greater

p63 abundance starts differentiation earlier to serve for the regeneration needs. While the other sister chromatid with lower guider histones and less p63 abundance will either differentiate or return to the stem cell fate depending on the regeneration needs. This model summarizes our current findings. We hope to further address this question in future studies, including experiments using the transplantation approach as this reviewer suggested.

Figure R1. Analysis of cell fate and proliferation of cultured HBCs with switching coating condition from fibronectin (FN) to tropoelastin (Tropo). (A) Illustration showing the experimental procedure. The HBCs were cultured for 1, 2, 3, or 5 days on FN coating, respectively, and then re-plated to Tropo coating dishes and cultured for 1.5 days until fixation and imaging. The bottom two panels indicate FN and Tropo coating conditions without switch. (B) Brightfield images of cultured HBCs in different coating conditions. Scale bar: 100 μ m. (C) Immunostaining of p63 and Ki67 in cultured HBCs. Scale bar: 40 μ m. (D) Quantification of p63 intensity in three different coating conditions. (E) Percentage of p63+ cells in three different coating conditions. Values represent average \pm SEM. n.s., no significance, * $p < 0.05$, **** $p < 0.0001$ by Mann-Whitney test.

The in vitro and in vivo experiments with nocodazole disruption of microtubule-based distribution are exciting, but the authors don't offer a way to assess off-target effects on the cells. That consideration of off-target effects is needed to conclude definitively that the disruption of asymmetry is the cause of the substantial differences in OE regeneration and olfactory-guided behavior in vivo. Other types of HBC manipulation – of Notch signaling and of Rac distribution after injury – also cause an alteration in the balance of types of HBC progeny (Louie et al, 2023; 2024). It is exciting to ponder

whether histone asymmetry is the ultimate means by which these diverse manipulations have their effect, but these alternatives need to be considered and controlled for.

Nocodazole (NZ), as a microtubule depolymerization drug, has been broadly used in cell cycle studies across various cell types. NZ treated cells under the used dose are arrested at prometaphase. However, NZ's effect on microtubule activity is reversible. The cells can proceed with mitosis when NZ is washed out in culture or metabolized *in vivo*. In our previous study, we established a system to transiently disrupt microtubule dynamics by NZ in *Drosophila* male germline. This treatment resulted in randomized sister chromatid segregation and altered histone inheritance in germline stem cells (Rajesh Ranjan et al. *Cell Stem Cell*. 2019. PMID: 31564548). In the current work, we first observed temporarily asymmetric microtubule dynamics in prophase to prometaphase HBCs (Supplementary Fig. 8A), similar to what we reported in *Drosophila* germline stem cells. This prompted us to use NZ treatment after HBC activation, in order to disrupt this microtubule asymmetry and test whether it would affect asymmetric histone inheritance in HBCs. Our results in mouse HBCs are consistent with those reported in *Drosophila* germline stem cells. We documented the effects of NZ treatment in cultured HBCs and found that they re-entered mitosis upon NZ washout (Supplementary Fig. 8B). In the OE, HBCs maintained high proliferative activity after a single dose of NZ administered during early OE regeneration (Fig. 6B). We therefore conclude that the impaired OE regeneration caused by disrupting asymmetric histone distribution was not an artifact of the NZ treatment. Of course, we cannot completely exclude off-target effects, as is the case with any pharmacological approach.

We appreciate the reviewer's insight and totally agree that it is interesting to determine whether histone asymmetry is the ultimate mechanism through which these diverse manipulations (e.g., alterations in Notch signaling, Rac activity, NZ treatment, etc.) have their effects. We have added a discussion to address this point. Future studies will be needed to explore this important question.

In addition, food-finding is a relatively crude behavioral measure of olfactory function. (What is a peanut bar, BTW? Perhaps provide a product name.) One wishes the authors had employed a second test to validate the result.

We agree with this reviewer's suggestion and have performed an additional behavioral assay, the olfactory preference test, following the protocol described in Rochelle M Witt et al. *JoVE*. 2009 (PMID: 19229182). The mice were subjected to the same treatments as those used for food seeking behavior. Specifically, the mice were subjected to 1 dose of Methimazole (MMZ) injection (50 mg/kg) and, 1 day after, 1 dose of NZ injection (15 mg/kg). Briefly, mice were tested individually. The mouse was habituated sequentially in three bedding-free clean cages (8 H x 10 W x 18 L, in inches) for 15 minutes each, followed by a final 15-minute habituation in the testing cage. A piece of filter paper (2 x 2 inches) containing ~0.5 mL of the test odorant solution was placed at one side of the cage, opposite to the mouse's initial position. For each trial, the tested mouse was

exposed to a new odorous filter paper for 3 minutes in a new cage. The total probe time (with the nose <1 mm from the filter paper) was quantified from the recorded videos by a blind observer. The concentrations of odorant solutions are listed below: lemon extract (1:10 dilution in distilled water); Vanilla extract (1:10 dilution in distilled water); Peanut butter (1 g per 10 mL odorless mineral oil). All cages and materials were thoroughly cleaned with 70% ethanol and dried between trials to prevent cross-contamination. The results showed that NZ treated mice spent significantly less time exploring the filter paper with attractive odors (Vanilla and Peanut butter) than non-NZ treated control mice on Day 15 post MMZ injection, while no significant difference was observed between the two groups on Day 28, suggesting that disruption of asymmetric histone inheritance in HBCs by NZ treatment leads to attenuated olfactory behavior recovery. We have included these results in the revised manuscript (Supplementary Fig. 9).

We have also added the information of the peanut bar in Methods.

Figure R2. (Figure S9) Olfactory preference test on Day 15 and Day 28 post-MMZ injection with NZ treatment. (A) Illustration of olfactory preference test in NZ-treated mice and non-NZ treated control mice. The mice were tested as a cohort, with each mouse exposed to an odor source for 3 minutes. The total probe time for each mouse was quantified. The cohort then proceeded to the next odor only after all mice in the cohort had completed the test for the current odor. $N_{Ctrl} = 8$ mice; $N_{NZ\ treat} = 10$ mice. (B) Quantification of total probe time from the olfactory preference test on Day 15 and Day 28 post-MMZ injection with NZ treatment. Total probe time in Ctrl mice on Day 15: Lemon = 3.88 ± 0.96 , Vanilla = 12.75 ± 2.22 , Peanut butter = 27.74 ± 8.27 ; Total probe time in NZ-treated mice on Day 15: Lemon = 3.19 ± 0.81 , Vanilla = 6.48 ± 1.35 , Peanut butter = 5.51 ± 1.50 . Total probe time in Ctrl mice on Day 28: Lemon = 2.00 ± 0.58 , Vanilla = 2.40 ± 0.41 , Peanut butter = 8.79 ± 2.65 ; Total probe time in NZ-treated mice on Day 28: Lemon = 1.14 ± 0.10 , Vanilla = 2.53 ± 0.42 , Peanut butter = 7.28 ± 2.51 . Values represent average \pm SEM. n.s., no significance, * $p < 0.05$ by Mann-Whitney test.

There are some deficiencies in the paper. The authors do a poor job of describing the state of the field in the introduction to the paper. The olfactory epithelium contains TWO stem cell populations: the normally dormant HBCs and a subset within the heterogeneous population of GBCs, which contribute to the continuously active generation of neurons under homeostatic and even accelerated neurogenesis. The latter has been established through functional assessments of tissue stemness and tentatively identified on the basis of multiple molecular markers. So that the readers who are not expert in olfactory biology do NOT come away with the misapprehension that HBCs are THE stem cell of the OE (which error, unfortunately, has not been eliminated from the literature).

We appreciate the reviewer's comments. In response, we have revised the manuscript to include the current view of olfactory stem cells in the OE, as follows: "... *This regenerative capacity is due to the presence of two stem cell populations in the OE, the horizontal basal cells (HBCs)^{2,3} and some subtypes of globose basal cells (GBCs)⁴. HBCs reside along the basement membrane, forming a single layer at the bottom of the OE. GBCs reside on the top of HBCs and are often organized into multiple layers^{2,5}.*" We didn't intend to overlook that some GBC subtypes function as stem cells. GBCs were not included in the initial submission primarily because our study focused on the HBC population.

In addition, there is a fundamental complexity to the response to lesion here that the authors do not address. At day 1 after methimazole injury p63 levels are essentially nil, which comports with previous observations using methyl bromide lesion (in which the intoxication is more time-locked and limited). As a consequence, all the HBCs should begin to activate from their dormancy. By looking at day 2 and focusing on cells that have some level of p63, the events they are studying are slightly different than pure activation and indicative of how context may be influencing the downstream events of activation. This is a subtle difference but may be highly relevant and/or specific to the form of lesion. Again, to ensure that the field comes away with a true understanding of how the regenerative process unfolds, these complexities need to be presented by the authors.

We used a single dose of methimazole to lesion the OE. The half-life of plasma methimazole is approximately 1.5 hours in mice (Fang Xie et al. *Drug Metab Dispos.* 2011. PMID: 21415250) and around 3 hours on average in humans (G G Skellern et al. *Br J Clin Pharmacol.* 1980. PMID: 7356900; J P Kampmann et al. *Clin Pharmacokinet.* 1981. PMID: 6172233; Y Okamura et al. *Endocrinol Jpn.* 1986. PMID: 3830069). Thus, the duration of methimazole action is well-controlled and time-constrained, although its pharmacokinetics in OE tissue remain unknown, as is also the case for methyl bromide.

As the reviewer noted, p63 levels are downregulated on Day 1 post both methimazole and methyl bromide treatment, indicating that HBCs exit quiescence and enter the cell cycle, i.e., they are activated. The reviewer also raised a valid point that different lesion

methods may activate HBCs through distinct mechanisms and temporal dynamics. However, our study mainly focused on the cell division process following stem cell activation. We appreciate the reviewer's insight and have added a brief discussion to highlight the potential complexities arising from different lesion methods.

Finally, we want to point out that telophase HBCs are rare in regenerating OE. To address this, we examined the proliferation dynamics of HBCs during early regeneration and found that their proliferation peaks on Day 2. This offers an optimal time window to identify more telophase HBCs for studying histone inheritance in this system.

There are a few minor problems with aligning text, figure legends, and figures. 1) First citation for Fig 1E and 1F in the text are inverted with respect to the labeling of the figure panels. 2) Figure legend S1 is wrong and does not correspond correctly with the panels in the Figure. 3) Please add labels for the markers in the combined image in Figure 3. 4) Also in Figure 3 – some cells connected by dotted lines are clearly in DIFFERENT clusters, despite what is said in the Figure Legend.

- 1) We thank the reviewer for pointing out this error and have corrected it in the revised manuscript.
- 2) Thank the reviewer for pointing this out. We have carefully checked that the legends now correspond correctly to the panels in the revised manuscript.
- 3) We have added color codes to the labels in Figure 3A.
- 4) We have double checked all sister chromatids connected by dotted lines in the Z-stack confocal images and confirmed that they are correctly labeled.

Reviewer #2

This study reveals that asymmetric histone inheritance regulates cell fate specification in mammalian adult stem cells during tissue regeneration. Using the mouse olfactory epithelium (OE) model, the authors demonstrate that histones H4, H3, and variant H3.3—but not H2A-H2B—are asymmetrically distributed in 30–40% of dividing horizontal basal cells (HBCs) during injury-induced regeneration. This asymmetry correlates with unequal partitioning of the transcription factor p63 and asynchronous transcription re-initiation (evidenced by asymmetric RNA Pol IIS2ph/S5ph), priming daughter cells for distinct fates. Functional disruption via nocodazole (NZ) abolishes histone/p63 asymmetry, impairs OE regeneration, and delays olfactory recovery, underscoring the essential role of epigenetic inheritance in tissue repair. Single-cell RNA sequencing of paired HBC daughters further confirms asymmetric multilineage priming, linking H3.3 transcript asymmetry to divergent differentiation trajectories. These findings establish histone asymmetry as a conserved mechanism guiding stem cell plasticity in mammals, with implications for regenerative therapies targeting epigenetic regulators. But this article lacks the molecular mechanism of asymmetric histone distribution. The

most critical step in histone inheritance during cell division is the redistribution of parental histones at the DNA replication fork during DNA replication.

We appreciate the reviewer's acknowledgment that our findings "establish histone asymmetry as a conserved mechanism guiding stem cell plasticity in mammals, with implications for regenerative therapies targeting epigenetic regulators". This is the first demonstration of asymmetric histone distribution and its functional significance in mammalian adult stem cells, indicating that this phenomenon is conserved across phyla.

We also agree that understanding how histone asymmetry is established is a critical question. In this study, we find that both replication-dependent histones (e.g., H3 and H4) and replication-independent histone variant H3.3 display asymmetry, indicating that replication as well as other processes such as transcription contribute to this phenomenon. A full understanding of these mechanisms is beyond the scope of a single study and will require future investigation.

Main comments:

1. Core Issue: Lack of Molecular Mechanism for Asymmetric Histone Segregation

The study demonstrates asymmetric histone (H4/H3/H3.3) inheritance in HBCs but fails to address how histones are directionally segregated during DNA replication—the most critical step for epigenetic inheritance. In mammalian cells: DNA replication initiates at multiple origins, with parental histones redistributed to both leading and lagging strands at each fork (Xu et al., Science 2010). Symmetry challenge: This bidirectional redistribution should theoretically equalize parental histone distribution between sister chromatids, making the observed asymmetry (30–40% of divisions) mechanistically puzzling. The authors neither characterize replication-coupled histone recycling nor provide evidence for biased histone transfer (e.g., strand-specific retention or chaperone-mediated sorting), leaving a gap between the observed asymmetry and established replication biology.

Although exploring the molecular mechanisms underlying asymmetric histone inheritance is undoubtedly important, as explained above, it is not the focus of the current study. Our work represents the first effort to understand how histone inheritance influences cell fate choices in mammalian adult stem cells. Mechanistic studies will require additional tools and substantial efforts beyond the scope of the current study. Indeed, even in *Drosophila*, more than a decade of work has yielded several critical yet still limited progresses, such as characterizing replication-coupled histone incorporation, given the small number of stem cells *in vivo* and the technical challenge of purifying them for genomic analyses. It is also worth noting that much of the "established replication biology" comes from studies in unicellular organisms like yeast or in cultured cells, where obtaining sufficient cell numbers for genomic studies is not a barrier.

Using *Drosophila* male germline stem cells as an *in vivo* model in a multicellular organism, it has been shown that strand-specific canonical histone incorporation and

biased replication fork movement can contribute to the establishment of histone asymmetry (Matthew Wooten et al. *Nat Struct Mol Biol.* 2019. PMID: 31358945). However, whether mammalian adult stem cells exhibit similar biased replication fork movement remains to be determined, although unidirectional fork movement has been reported in mammalian cells (Torsten Krude et al. *J Cell Sci.* 2009. PMID: 19657016; Ronald Lebofsky et al. *Mol Cell Biol.* 2005. PMID: 16024811; Slavica Stanojic et al. *Sci Rep.* 2016. PMID: 26976742; Kathrin Marheineke et al. *Nucleic Acids Res.* 2005. PMID: 16332696; Avital Zerbib et al. *Int J Mol Sci.* 2023. PMID: 37298562). At present, we don't have the necessary tools to address this question in mammalian systems, such as cell-type- and stage-specific drivers for labeling chromatin fibers.

Xu et al. *Science.* 2010 (PMID: 20360108) showed DNA replication-dependent histone deposition in Hela cells, which typically undergo symmetric divisions without cell fate changes.

2. The claimed asymmetry conflicts with prior work showing: equal histone partitioning: Canonical histones (H3-H4)₂ are randomly deposited on both daughter strands during replication (Petryk et al., *Science* 2018). (H3.3-H4)₂ are symmetrically deposited on both daughter strands during replication, as well (Xu et al., *Nature communications* 2022). H3.3 dynamics: While H3.3 is replication-independent, its asymmetric inheritance (Fig. 2E) lacks mechanistic explanation. The study does not distinguish whether asymmetry arises from de novo deposition (post-replication) or replication-coupled recycling.

As explained above, there are fundamental differences between the systems we use (i.e., *in vivo* adult stem cells) and those systems cited here (i.e., Nataliya Petryk et al. *Science.* 2018. PMID: 30115746; Xiaowei Xu et al. *Nat Commun.* 2022. PMID: 35523900), where the authors used cultured mouse embryonic stem cells normally undergoing symmetric cell divisions.

H3.3 incorporation is replication-independent but transcription-dependent. In telophase HBCs, we found that sister chromatids inheriting asymmetric H3.3 differentially re-initiate transcriptional activity (Fig. 3 in the revised manuscript), highlighting the biological significance of this asymmetry. However, as noted above, fully elucidating the underlying mechanisms is beyond the scope of a single study and will require future investigation.

3. To reconcile their findings with replication biology, the authors should: Track histone dynamics during S-phase: Use pulse-chase labeling (e.g., Click-IT EdU combined with H3-H4 immunoprecipitation) to map parental histone redistribution at replication forks in HBCs. Compare histone retention on leading vs. lagging strands via eSPAN. Validate in replication mutants: Perturb replication symmetry (e.g., MCM2 mutants that bias histone retention) and assess asymmetry persistence.

As noted above, there are fundamental biological differences between the system we use (*in vivo* adult stem cells) and the systems typically used in the “replication biology” field. Therefore, it is not appropriate to “reconcile” our findings with those studies, as they address distinct biological questions: How asymmetric histone inheritance associates and may regulate cell fate changes *in vivo* versus how symmetric histone inheritance maintains cell identity.

We took enormous efforts trying to perform an assay to examine the “old” and “new” histone distribution using HaloTag labeling in S-phase cultured HBCs. Prior to Halo ligand labeling, we first validated that G1 synchronization did not affect cell fates or division modes in cultured HBCs (Fig. R3A-E). We then synchronized the cultured HBCs in early G1 and performed the first Halo ligand labeling to mark old histone (red fluorescence). After releasing the cells with mevalonate for 4 hours, we applied the second Halo ligand to label new histone (green fluorescence) for 2 or 4 hours. Using a short EdU pulse (20 minutes) to mark active S phase, we found that old and new H3 exhibited both non-overlapping distribution patterns (colocalization index ≤ 0.69 , N=13, 27.1%) and overlapping patterns (colocalization index > 0.69 , N=35, 72.9%), suggesting that biased histone incorporation can occur during DNA replication (Fig. R3F-H). We previously reported that asymmetrically dividing stem cells exhibit greater colocalization of EdU with newly synthesized histones during S phase, likely resulting from the spatially distinct incorporation of old *versus* new histones (Brendon E M Davis et al. *Sci Adv.* 2025. PMID: 40020063). Consistently, we found that culture HBCs exhibited higher colocalization of EdU with new H3 on average, likely because old and new H3 were differentially incorporated in asymmetrically dividing HBCs (Fig. R3I). Therefore, the average colocalization index of new H3 with EdU (0.69) was used as a cutoff to define overlapping *versus* non-overlapping distribution patterns. (Fig. R3H-I). However, due to the preliminary nature of this piece of data, we decided not to include it in the revised manuscript and will revisit them in a future study with more comprehensive analyses.

Notably, in this protocol, we used a clearance buffer to remove unbound histones (Brendon E M Davis et al. *Sci Adv.* 2025. PMID: 40020063). Co-localization analysis was performed primarily in early to mid-S phase HBCs, identified by uniform EdU staining across the nucleus. Nuclei displaying punctate EdU patterns, indicative of late S phase, were excluded to minimize potential bias toward non-overlapping patterns between old and new histones arising from the late-replicating nature of heterochromatin. These results are largely consistent with the percentage of asymmetric histone distribution observed using both the tagged method and immunostaining (Fig. 1-2 in the revised manuscript).

Figure R3. The distribution of “old” and “new” histone in replicating HBCs. (A) Immunostaining of p63 and EdU labeling (20 mins) in lovastatin (Lovas) treated HBCs (48 hrs) and non-treated control (Ctrl) HBCs. **(B)** Quantification of p63 levels. Ctrl p63 level: 119732 ± 4679 ; Lovas-treated p63 level: 113906 ± 5639 . $p = 0.109$ by Mann-Whitney test. **(C)**

Quantification of EdU+ cells in p63+ HBCs. $N_{\text{Ctrl}} = 214$; $N_{\text{Lovas treat}} = 431$. n.s. and ** $P < 0.01$ by Mann-Whitney test. **(D)** Distribution patterns of p63 in telophase HBCs with lovastatin treat (48 hrs) and release (24 hrs). **(E)** Quantification of p63 distribution patterns in telophase HBCs ($N = 35$). **(F)** Illustrations showing the cell cycle information of culture HBCs on FN coat (left) and the experimental procedure of early G1 synchronization and release with old and new histone labeling (right). A short EdU pulse labels the S phase cells. **(G)** Representative images of old (red) and new (green) histone H3 distribution patterns in an early-to-mid S phase HBC (identified by the EdU patterns in magenta). **(H)** Quantification of the colocalization index between old H3 and new H3 signals in early-to-mid S phase HBCs ($N = 48$). Cutoff of non-overlapping = 0.69, as the red dotted line indicated. **(I)** Difference between colocalizations of new H3 with EdU and old H3 with EdU in early-to-mid S phase HBCs nuclei ($N = 34$). New H3 vs EdU: 0.69 ± 0.03 ; Old H3 vs EdU: 0.61 ± 0.03 . * $P < 0.05$ by Mann-Whitney test. All values represent average \pm SEM. Scale bars: 50 μm in **(A)**; 10 μm in **(D)** and **(G)**.

Further comparison of histone retention using eSPAN could provide valuable insight into replication-dependent histone incorporation. However, the eSPAN technique, whether applied to wild-type or mutant cells, requires millions of asymmetrically dividing cells to profile parental histone redistribution at replication forks in HBCs. Currently, no method exists to isolate these asymmetrically dividing cells for genomic assays.

4. If replication-independent mechanisms drive asymmetry, the authors must: Rule out technical artifacts: Overexpressed H4-mScarlet/H3.3-mScarlet (Fig. 1D, 2E) may disrupt native chromatin recycling. Endogenous tagging (e.g., C-terminal tags) is needed. Clarify temporal control: Asymmetry might emerge post-replication, but this is not addressed.

As noted above, the mechanisms driving histone asymmetry are not fully understood and may include both replication-independent and replication-dependent processes in this system. In this study, we report the phenomenon of histone and histone variant asymmetry in adult mammalian cells and demonstrate its biological significance. Elucidating the underlying mechanisms will be the focus of future studies.

We agree with the reviewer that endogenous tagging would be a more ideal approach for tracing histone distribution. However, the only reported case of endogenous histone tagging in mammalian system is the H3-tagged iCOUNT mouse strain (Annina Denoth-Lippuner et al. *Cell Stem Cell*. 2021. PMID: 34525348). We carefully analyzed this mouse line obtained from the Jackson Laboratory (JAX #037318) but found no detectable tagged H3 signal in cells from multiple tissues, including the OE.

We thus adopted the inducible H4-mScarlet and H3.3-mScarlet mouse strains for our histone distribution studies.

5. related to 4. Overexpressed H4-mScarlet cannot reflect the exact distribution of endogenous H4. The chromatin of the cell who acquires more H3.3-H4 tends to transcribe and be looser, then the density of nucleosomes should be lower. How could the cell acquire more H4? The explanation of these results could not make sense.

Reviewer #1 has a similar concern, and we have addressed this concern as the following, “We acknowledge the reviewer’s concern about the use of fluorescently tagged H4 in a transgenic mouse line. Previous studies have shown that the C-terminally tagged-H4 recapitulates endogenous H4 patterns assessed by immunostaining and chromatin immunoprecipitation (Hiroshi Kimura et al. *J Cell Biol.* 2001. PMID: 11425866; Kami Ahmad et al. *Molecular Cell.* 2002. PMID: 12086617), indicating that the tag does not affect H4 localization, cell cycle kinetics and replication-dependent chromatin incorporation. In our study, the expression of the tagged-H4 was induced by Doxycycline, and we titrated the Doxycycline dose and used a low dose to achieve a detectable but low level of tagged-H4 expression. Our Western blot result showed that tagged-H4 is expressed ~6% of the endogenous H4 levels (Fig. 1F and Supplementary Fig. 1B in the revised manuscript). Importantly, we compared the HBCs dynamics during OE regeneration and found no difference between H4mS (Ctrl, in which no tagged-H4 expression) and H4mS;rtTA (the tagged-H4 is expressed due to the presence of rtTA) (Supplementary Fig. 2A-C in the revised manuscript). Furthermore, our results of using H4 antibody staining in culture HBCs (Fig. 2C in the revised manuscript) showed similar pattern and percentage of asymmetric H4 distribution to that in telophase HBCs expressing tagged H4 *in vivo*. Thus, we conclude that the induced expression of tagged-H4 does not affect the H4 biology in our system. Notably, a key advantage of using fluorescently tagged H4 rather than antibodies in immunostaining is that it eliminates concerns about antigen accessibility, which can be an issue in mitotic cells when chromosomes are highly condensed.”.

Because H4 does not have known variants, overall H4 levels should reflect overall nucleosomal density. However, nucleosome composition and associated post-translational modifications (PTMs) also influence transcriptional activity in a locus-specific manner. For example, H3.3-H4-containing nucleosomes typically correlate with actively transcribed regions, whereas H3-H4-containing nucleosomes can be associated with either active or repressive states depending on their PTMs (e.g., H3K4me3 and H3K36me3 mark active regions, while H3K9me3 and H4K20me3 are linked to gene silencing). Additional sequencing-based approaches will be needed to fully characterize how these chromatin features contribute to differential gene expression in the two daughter cells arising from HBC division. In our initial effort, we performed single-cell RNA-seq of paired daughters and confirmed asymmetric multilineage priming, linking H3.3 transcript asymmetry to divergent differentiation trajectories. We have expanded this discussion in the revised manuscript to clarify these points.

6. Technologically, how did the author ensure that the two daughter cells in the tissues are in the same horizontal plane while preparing the slices? If the focus of the two daughter cells is not in the same plane when taking pictures, the one out of focus looks like with a weak signal, resulting in the observed asymmetry signal of detected proteins. But this is not the fact; it is only a systematic error.

We used the following criteria when quantifying the nuclear (sister chromatids) fluorescent signals in telophase HBCs from confocal images with Z stacks to ensure both nuclear (sister chromatids) are entirely covered. Specifically, we first examined both sister chromatids by confocal scanning and then selected telophase HBCs with intact nuclei for imaging. All signal quantifications, whether from fluorescent tags or immunostaining, were performed using 3D reconstructions, measuring total histone amount across the entire nucleus or set of sister chromatids rather than from a single slice. Therefore, our quantification does not require the two daughter cells to be in the same focal plane. We have clarified this throughout the main text, figure legends, and Materials & Methods in the revised manuscript. Moreover, our *in vivo* findings are further corroborated by our results obtained from cultured cells.

7. Figure 5B, is the division of cells affected by NZ treat? NZ is usually used as a G2 cell cycle blocker. The phenotype should rule out the cell cycle arrest caused by NZ.

We acknowledge the reviewer's concern. Reviewer #1 has a similar concern, and we have addressed this concern as the following, "Nocodazole (NZ), as a microtubule depolymerization drug, has been broadly used in cell cycle studies across various cell types. NZ treated cells under the used dose are arrested at prometaphase. However, NZ's effect on microtubule activity is reversible. The cells can proceed with mitosis when NZ is washed out in culture or metabolized *in vivo*. In our previous study, we established a system to transiently disrupt microtubule dynamics by NZ in *Drosophila* male germline. This treatment resulted in randomized sister chromatid segregation and altered histone inheritance in germline stem cells (Rajesh Ranjan et al. *Cell Stem Cell*. 2019. PMID: 31564548). In the current work, we first observed temporarily asymmetric microtubule dynamics in prophase to prometaphase HBCs (Supplementary Fig. 8A), similar to what we reported in *Drosophila* germline stem cells. This prompted us to use NZ treatment after HBC activation, in order to disrupt this microtubule asymmetry and test whether it would affect asymmetric histone inheritance in HBCs. Our results in mouse HBCs are consistent with those reported in *Drosophila* germline stem cells. We documented the effects of NZ treatment in cultured HBCs and found that they re-entered mitosis upon NZ washout (Supplementary Fig. 8B). In the OE, HBCs maintained high proliferative activity after a single dose of NZ administered during early OE regeneration (Fig. 6B). We therefore conclude that the impaired OE regeneration caused by disrupting asymmetric histone distribution was not an artifact of the NZ treatment. Of course, we cannot completely exclude off-target effects, as is the case with any pharmacological approach."

Reviewer #3

This is an interesting study. The authors show the biological significance of epigenetic histone inheritance and conservation of asymmetric histone inheritance in asymmetrically dividing stem cells. But I still have a few questions:

1. In the Tet-inducible H4-tag transgenic mouse model, H4-mScarlet is overexpressed or knock-in in situ? Some researchers do not accept the results from overexpressed histone tag, because their different characteristics with the endogenous histone.

Reviewers #1 and #2 have a similar concern, and we have addressed this concern as the following, “We acknowledge the reviewer’s concern about the use of fluorescently tagged H4 in a transgenic mouse line. Previous studies have shown that the C-terminally tagged-H4 recapitulates endogenous H4 patterns assessed by immunostaining and chromatin immunoprecipitation (Hiroshi Kimura et al. *J Cell Biol.* 2001. PMID: 11425866; Kami Ahmad et al. *Molecular Cell.* 2002. PMID: 12086617), indicating that the tag does not affect H4 localization, cell cycle kinetics and replication-dependent chromatin incorporation. In our study, the expression of the tagged-H4 was induced by Doxycycline, and we titrated the Doxycycline dose and used a low dose to achieve a detectable but low level of tagged-H4 expression. Our Western blot result showed that tagged-H4 is expressed ~6% of the endogenous H4 levels (Fig. 1F and Supplementary Fig. 1B in the revised manuscript). Importantly, we compared the HBCs dynamics during OE regeneration and found no difference between H4mS (Ctrl, in which no tagged-H4 expression) and H4mS;rtTA (the tagged-H4 is expressed due to the presence of rtTA) (Supplementary Fig. 2A-C in the revised manuscript). Furthermore, our results of using H4 antibody staining in culture HBCs (Fig. 2C in the revised manuscript) showed similar pattern and percentage of asymmetric H4 distribution to that in telophase HBCs expressing tagged H4 *in vivo*. Thus, we conclude that the induced expression of tagged-H4 does not affect the H4 biology in our system. Notably, a key advantage of using fluorescently tagged H4 rather than antibodies in immunostaining is that it eliminates concerns about antigen accessibility, which can be an issue in mitotic cells when chromosomes are highly condensed.”.

2. The authors mentioned that histone H4, especially (H3.3-H4)₂ tetramers and SUV39H1 display asymmetric inheritance in HBCs. Does the (H3-H4) tetramers with specific modification, such as H3K27me₃, H3K9me₃, H3K27ac, display more significant asymmetric inheritance?

We performed the immunostaining for H3K27me₃, H3K9me₃, H3K27ac in cultured HBCs. The staining of H3K9me₃ and H3K27ac showed obvious border effects, probably due to the uneven antibody penetration, which makes the quantification unreliable. H3K27me₃ staining showed no border effects (Fig. R4A). Preliminary analysis of H3K27me₃ and p63 levels showed that H3K27me₃ distribution patterns correlate with p63 distribution patterns in telophase HBCs (Fig. R4B). This result suggests that H3K27me₃ may contribute to asymmetric histone distribution and cell fate determination in olfactory HBCs. Due to the preliminary and out-of-scope nature of this piece of data, we decided not to include it in the revised manuscript.

Figure R4. The distribution patterns of histone modifications in cultured HBCs. (A) Immunostaining of H3K9me3, H3K27ac and H3K27me3 along with p63 in cultured HBC colonies. Scale bar \square 50 μ m. **(B)** Example of asymmetric (top panel) and symmetric (bottom panel) distribution patterns of H3K27me3 and p63 in telophase HBCs. Scale bar \square 10 μ m.

3. The authors mentioned that asymmetric mitotic spindle activity is required for asymmetric histone inheritance. Does DNA replication-coupled histone recycling participate in this process? eSPAN or SCAR-seq in the in-vitro cell model could explain this well.

Reviewer #2 has a similar concern, and we have addressed this concern as the following, "Further comparison of histone retention using eSPAN could provide valuable insight into replication-dependent histone incorporation. However, the eSPAN technique, whether applied to wild-type or mutant cells, requires millions of asymmetrically dividing cells to profile parental histone redistribution at replication forks in HBCs. Currently, no method exists to isolate these asymmetrically dividing cells for genomic assays."

4. It is somewhat difficult to understand that p63 high to maintain the quiescence of HBCs, Meanwhile, p63 high cells own higher H3.3-H4 tetramers and preferentially express the differentiation-related gene (scrNA-seq results). The author should explain this more clearly.

It has been reported that p63 helps to maintain the quiescent state of HBCs (Adam Packard et al. *J Neurosci*. 2011. PMID: 21677159; Nikolai Schnittke et al. *PNAS*. 2015. PMID: 26305958). Consistent with this, our data and others' studies show that p63 downregulation following acute injury triggers HBCs activation and subsequent differentiation for tissue repair.

We thank the reviewer for raising this very interesting question. We share the same curiosity regarding the role of p63 during mitosis. The scRNA-seq data were from post-mitotic cells, where the daughter cell carrying higher H3.3 levels preferentially expressed the differentiation-related genes, but showed lower p63 levels. In telophase HBCs, however, higher p63 was accompanied by higher H3.3 levels. In the Discussion, we proposed a model that delineates how histones and p63 may work together in a sequential order to influence HBC fate determination, as summarized here. *“During late telophase, when the chromosomes start de-condensation, the guider histones (e.g., H4, H3, and H3.3) were asymmetrically distributed between two sets of sister chromatids (Fig. 2C-E in the revised manuscript). The sister chromatid inheriting higher guider histones and greater p63 abundance initiates the transcription process sooner, which may allow this future daughter cell to reach a transition stage prepared for differentiation earlier. The other sister chromatid inheriting lower guider histones and less p63 abundance initiates transcription later, which may delay its future daughter to reach this transition stage. After mitosis, the daughter cell inheriting more guider histones will start differentiation toward a progenitor or a sustentacular cell sooner for prioritizing regeneration needs, which is consistent with the observation of high histone levels in differentiating progenitors (p63-negative cells, Fig. 1E). The other daughter cell inheriting less guider histones will serve for replenishing the stem cell pool or differentiation depending on the sufficiency of the earlier-generated progenitors for the tissue repair.”*

5. The authors disrupted the asymmetric histone inheritance by nocodazole treatment in mice and observed that the recovery in the OE thickness is slower than the control. How to exclude the effect of cell cycle arrest caused by nocodazole?

We acknowledge the reviewer’s concern. Reviewers #1 and #2 have a similar concern, and we have addressed this concern as the following, “Nocodazole (NZ), as a microtubule depolymerization drug, has been broadly used in cell cycle studies across various cell types. NZ treated cells under the used dose are arrested at prometaphase. However, NZ’s effect on microtubule activity is reversible. The cells can proceed with mitosis when NZ is washed out in culture or metabolized *in vivo*. In our previous study, we established a system to transiently disrupt microtubule dynamics by NZ in *Drosophila* male germline. This treatment resulted in randomized sister chromatid segregation and altered histone inheritance in germline stem cells (Rajesh Ranjan et al. *Cell Stem Cell*. 2019. PMID: 31564548). In the current work, we first observed temporarily asymmetric microtubule dynamics in prophase to prometaphase HBCs (Supplementary Fig. 8A), similar to what we reported in *Drosophila* germline stem cells. This prompted us to use NZ treatment after HBC activation, in order to disrupt this microtubule asymmetry and test whether it would affect asymmetric histone inheritance in HBCs. Our results in mouse HBCs are consistent with those reported in *Drosophila* germline stem cells. We documented the effects of NZ treatment in cultured HBCs and found that they re-entered mitosis upon NZ washout (Supplementary Fig. 8B). In the OE, HBCs maintained high proliferative activity after a single dose of NZ administered

during early OE regeneration (Fig. 6B). We therefore conclude that the impaired OE regeneration caused by disrupting asymmetric histone distribution was not an artifact of the NZ treatment. Of course, we cannot completely exclude off-target effects, as is the case with any pharmacological approach.”.

Minor:

1. The asymmetric of Pol IIS2ph distribution in the two daughter cells is not obvious.

The asymmetric distribution of Pol IIS2ph is now shown in Fig. 3D and quantified in Fig. 3D' with p63 in the revised manuscript.

2. The EU signal does not coincide with the Pol IIS2ph signal, please explain.

EU labels the newly synthesized RNAs. Since rRNAs account for 80-90% of total RNA and are transcribed by RNA Polymerase I, the EU signal tends to be intensively enriched in the nucleoli, unlike the Pol IIS2ph signal, which reflects transcription of mRNAs. However, quantification of the signals showed that both EU and Pol IIS2ph levels positively correlate with p63, indicating a positive association among them (Fig. 3D-F in the revised manuscript).

Reviewer #4

This manuscript investigates asymmetric histone inheritance in the context of adult mammalian stem cell fate specification. The authors identify that in mouse olfactory epithelial horizontal basal cells (HBCs), canonical histones H3 and H4, as well as the variant H3.3, are inherited asymmetrically during regeneration. This asymmetry is associated with differential p63 expression and asynchronous transcriptional re-initiation between daughter cells. Functional perturbation using nocodazole shows that disrupting histone asymmetry impairs olfactory epithelium (OE) regeneration. These findings suggest a novel role of histone inheritance in tissue regeneration. There are some issues regarding clarity and quality controls that need to be addressed.

We appreciate the reviewer's acknowledgment that our findings “suggest a novel role of histone inheritance in tissue regeneration”.

Major concerns:

The citations of Figures 1E and 1F are wrong.

We thank the reviewer for pointing this out and apologize for the oversight. We have corrected the error in the revised manuscript.

The conclusion that “the fusion protein accounts for approximately 10% of endogenous H4” from the Western blotting of Figure 1E is not convincing. Quantification of the WB from three habituation and a serial dilution of the samples to compare the amount of proteins is needed.

We thank the reviewer’s advice and have performed extra Western blotting analysis with three biological replicates, each with a serial dilution of the samples. The tagged H4 from each mouse OE tissue accounts for 5.6%, 6.3% and 5.3% of endogenous H4. On average, the fusion protein accounts for ~6% of endogenous H4 based on this assessment, which is less than our previous assessment of 10% based on one sample. We have included this new result in Fig. 1F (Sample 3) and Supplementary Fig. 1B (all blots) in the revised manuscript.

Figure R5. (Figure 1F and S1B) Immunoblot of H4mS;rtTA OE lysate on Day 2 post-MMZ injection with anti-H4 antibody. (A) The protein lysate was extracted from each of three mice (Sample 1 to Sample 3). The three lanes in each sample represent the original OE lysate, 1:2 and 1:4 dilution of the original lysate, respectively. Top band, H4-mScarlet fusion protein, ~40 KD; Bottom band, endogenous H4 protein, ~11 KD. **(B)** Quantification of H4-mScarlet fusion

protein compared to endogenous H4 protein of three biological duplicates. Note: Due to the low signals of 1:4 dilution lane, the first two lanes were used for quantification. H4-mScarlet amount is ~6% of the endogenous H4.

Figure S2B-C, what does the “control” mean? Please specify the condition. In addition, an explanation of the mScarlet signal in controls needs to be addressed.

The control in Supplementary Fig. 2 is the H4-mScarlet (H4mS) mouse line, which lacks the rtTA allele, as indicated in the figure legend. To avoid confusion, we have changed the label from “Ctrl” to “H4mS” in Supplementary Fig. 2 in the revised manuscript.

Expression of H4-mScarlet is not present in the absence of DOX administration. The weak and sparse red fluorescence signal observed in the H4mS control reflects autofluorescence from dying olfactory tissue or cells, which was not nuclear and therefore not included in all analyses.

Figure 1G and others, please add the line to the cells to indicate how the intensity is measured.

We thank the reviewer for this suggestion. We have replotted the p63/H4-mScar merged images and added a line indicating the path used to generate the line-plot quantification.

The Pol II S2p and S5p are identified to be asymmetrically distributed. How about the total amount of Pol II?

We performed immunostaining for Pan-Pol II in cultured HBCs. The data show that Pan-Pol II exhibits an overall symmetric distribution pattern in telophase HBCs, regardless of whether p63 distribution is asymmetric or symmetric (Fig. R6). These results suggest that unphosphorylated RNA polymerases are evenly distributed, whereas transcription is differentially re-initiated by RNA Pol II S5p and elongated by S2p. We have included this data in the revised manuscript as Supplementary Fig. 6B-D.

Figure R6. (Figure. S6B-D) The distribution patterns of Pan-RNA polymerase in telophase HBCs. (A) Immunostaining of unphosphorylated Pol II (Pan-Pol II) and p63 in telophase HBCs. Top panel, symmetric distribution pattern of Pan-Pol II in a telophase HBC with asymmetric p63 distribution. Bottom panel, asymmetric distribution pattern of Pan-Pol II in a telophase HBC with asymmetric p63 distribution. Scale bar: 5 μ m. **(B)** Quantification of the ratio of Pan-Pol II and p63 in telophase HBCs (N = 32). The cutoff for p63 and Pan-Pol II asymmetry is 1.5. **(C)** The correlation plot of ratios of Pan-Pol II and p63 in each telophase HBC. R = 0.522.

The description of Figure 4B doesn't match the Figure.

We apologize for this inconsistency and have revised the text to correspond with the figure.

For the scRNA-seq, quality controls need to be presented, and how to filter the low-quality cells. How many batches of this scRNA-seq are performed, as the scRNA-seq needs biological replicates?

We have added the following description for quality control and cell filtering in Method. "After aligning all scRNA-seq reads from 96 single cells (48 pairs) to the reference genome, the total number of genes detected per cell was examined and only those cells

with more than 10,000 detectable genes were retained for downstream analysis. Additionally, cells with more than 5% of reads mapping to mitochondrial genes were excluded from further analysis.”

All 96 cells presented in the manuscript met the quality criteria. On average, 1.3×10^4 genes were detected per cell (left panel of Figure R7), with approximately 2% of reads mapping to mitochondrial genes (right panel of Figure R7). Each single cell was treated as a separate biological replicate, and 96 independent RNA-seq libraries were prepared. We have included this data in Supplementary Fig. 7A in the revised manuscript.

Figure R7. (Supplementary Figure. 7A) Combined violin and box plots showing the total number of genes detected (left) and the mitochondrial gene ratio (right) in the scRNA-seq analysis. Each dot represents data from a single cell. N = 96.

In the scRNA-seq, does the expression of p63 follow the asymmetric distribution as detected in other figures?

We examined p63 expression in paired daughter cells and found that 35 pairs (72.9%) exhibited asymmetric cell fate specification based on p63 transcript levels (Fig. R8). This proportion is higher than the asymmetry observed at the protein level in telophase HBCs using immunostaining, likely reflecting increased transcription from mitosis to post-mitotic cells. We have included this data in Supplementary Fig. 7E in the revised manuscript.

Figure R8. (Supplementary Figure. 7E) Ratios of p63 transcripts between paired daughter cells based on scRNA-seq. Lower p63-expressing daughter cell normalized to 1; asymmetry defined as ≥ 1.5 , consistent with the methods quantifying immunostaining and tagged-histone datasets.

Figure 4H, “The other daughter cell carrying lower H3.3 levels show tendency either for self-renewal (higher p63, Trp53inp1, Myc) or for differentiation (lower Krt5, Krt14, Icam1) (Fig. 4H).” I’m confused that in previous results, H3.3 is high in P63 high cells, but why is the correlation opposite in scRNA-seq?

The statement that “H3.3 is high in p63-high cells” was based on imaging data collected from telophase HBCs, when chromosomes begin to decondense. The scRNA-seq data, however, were obtained from paired daughter cells immediately after the division, representing a slightly later time point likely in G1 phase.

In the revised manuscript, we propose a model (illustrated in Fig. 6G) that delineates how histones and p63 may work together in a sequential order to influence HBC fate determination, as summarized here. “During late telophase, when the chromosomes start de-condensation, the guider histones (e.g., H4, H3, and H3.3) were asymmetrically distributed between two sets of sister chromatids (Fig. 2C-E in the revised manuscript). The sister chromatid inheriting higher guider histones and greater p63 abundance initiates the transcription process sooner, which may allow this future daughter cell to reach a transition stage prepared for differentiation earlier. The other sister chromatid inheriting lower guider histones and less p63 abundance initiates transcription later, which may delay its future daughter to reach this transition stage. After mitosis, the daughter cell inheriting more guider histones will start differentiation toward a progenitor or a sustentacular cell sooner for prioritizing regeneration needs, which is consistent with the observation of high histone levels in differentiating progenitors (p63-negative cells, Fig. 1E). The other daughter cell inheriting less guider histones will serve to replenish the stem cell pool or differentiation depending on the sufficiency of the earlier-generated progenitors for the tissue repair. The post-mitotic transcriptomes of paired daughter HBCs are consistent with the output of differential cell fate priming due to asymmetric histone inheritance, where the daughter cell carrying higher H3.3 levels

preferentially expressed the differentiation-related genes, while the other daughter cell carrying lower H3.3 levels showed no preference for self-renewal or differentiation (Fig. 4H in the revised manuscript”).

Figure 4I, from the PCA plot, these renewed and activated HBC do not separate into two clusters as indicated in the results section.

Per the reviewer’s suggestion, we have added cluster boundaries for the two HBC populations in the PCA plot to more clearly depict the separation between Activated and Renewed HBCs.

It is important to note, however, that these two clusters remain transcriptionally close, reflecting their biological similarity immediately after division. This analysis has been incorporated into the revised manuscript as Fig. 4I.

Figure R9. (Figure. 4I) PCA plot of scRNA-seq data for all Renewed and Activated HBCs. Dashed lines denote boundaries; “x” marks the center of each cluster.

Figure 4L, quantification is needed to show the SUV39 staining.

We have now included the quantification of SUV39H1 distribution pattern in Supplementary Fig. 7I in the revised manuscript. The figure legend of Fig. 4L has been revised for clarity: “Asymmetric (31%) and symmetric (69%) SUV39H1 distribution in p63+ telophase HBCs (N=29) from primary culture. p63 shows the same distribution pattern as SUV39H1.”

Is CenH3 asymmetrically distributed in the scRNA-seq results?

We have examined CenH3 expression in paired daughter cells, as shown below in Fig. R10. Using a 1.5-fold ratio as the cutoff, the majority of pairs (33/48; 68.75%) showed asymmetric distribution.

Figure R10. CenH3 transcript levels in paired daughter cells from scRNA-seq analysis. The daughter cell with the lower Cenpa expression level was normalized to 1. A 1.5-fold threshold was used to define asymmetry, in line with the methods quantifying immunostaining and tagged-histone datasets. staining and tagged histone datasets.

The finding that H3.3 is asymmetrically inherited in contrast to *Drosophila* is intriguing. Could the authors expand on whether this is due to P63 or another mechanism that leads to this difference?

The reviewer raised an important point. Indeed, the asymmetric H3.3 distribution in mouse HBCs is intriguing. In *Drosophila*, previous studies have examined old versus new H3.3 inheritance patterns in male germline stem cells (Vuong Tran et al. *Science*. 2012. PMID: 23118191; Chinmayi Chandrasekhara et al. *PLoS Biol*. 2023. PMID: 37126497), which display largely symmetric patterns. However, the total levels of H3.3 have not been assessed as in the present work, nor has it been investigated whether the two sister chromatids differentially re-initiate the transcriptional activity. We have added this point in the revised Discussion.

In response to Reviewer #2, we have provided an explanation for H3.3 asymmetry in mammals: “H3.3 incorporation is replication-independent but transcription-dependent. In telophase HBCs, we found that sister chromatids inheriting asymmetric H3.3 differentially re-initiate transcriptional activity (Fig. 3), highlighting the biological significance of this asymmetry. However, as noted above, fully elucidating the underlying mechanisms is beyond the scope of a single study and will require future investigation.”.

While nocodazole treatment supports a causal link, it affects microtubule dynamics broadly. Is there a way to disentangle histone partitioning from spindle orientation more

specifically? Are there alternative approaches to target chromatin directly, like the authors did before, to mutate the phosphorylation sites in H3?

We acknowledge the reviewer's concern. Reviewers #1, #2 and #3 have a similar concern, and we have addressed this concern as the following, "Nocodazole (NZ), as a microtubule depolymerization drug, has been broadly used in cell cycle studies across various cell types. NZ treated cells under the used dose are arrested at prometaphase. However, NZ's effect on microtubule activity is reversible. The cells can proceed with mitosis when NZ is washed out in culture or metabolized *in vivo*. In our previous study, we established a system to transiently disrupt microtubule dynamics by NZ in *Drosophila* male germline. This treatment resulted in randomized sister chromatid segregation and altered histone inheritance in germline stem cells (Rajesh Ranjan et al. *Cell Stem Cell*. 2019. PMID: 31564548). In the current work, we first observed temporarily asymmetric microtubule dynamics in prophase to prometaphase HBCs (Supplementary Fig. 8A), similar to what we reported in *Drosophila* germline stem cells. This prompted us to use NZ treatment after HBC activation, in order to disrupt this microtubule asymmetry and test whether it would affect asymmetric histone inheritance in HBCs. Our results in mouse HBCs are consistent with those reported in *Drosophila* germline stem cells. We documented the effects of NZ treatment in cultured HBCs and found that they re-entered mitosis upon NZ washout (Supplementary Fig. 8B). In the OE, HBCs maintained high proliferative activity after a single dose of NZ administered during early OE regeneration (Fig. 6B). We therefore conclude that the impaired OE regeneration caused by disrupting asymmetric histone distribution was not an artifact of the NZ treatment. Of course, we cannot completely exclude off-target effects, as is the case with any pharmacological approach."

As the reviewer mentioned, we previously reported that in *Drosophila*, expression of H3 with threonine 3 to unphosphorylatable residue alanine (H3T3A) mutation disrupts asymmetric histone inheritance in germline stem cells and leads to germ cell loss and germline tumors. We made efforts to generate a viral vector with H3T3A mutation, and infected cultured HBCs derived from H3.3mS;rtTA and WT mice, and accessed distribution patterns of histone H3.3, Pol IIS2ph/Pol IIS5ph, and p63. The results showed that H3T3A mutation disrupted asymmetric H3.3 distribution, asynchronous transcription re-initiation and asymmetric p63 distribution in H3T3A+ telophase HBCs (Fig. R11), suggesting that targeting chromatin directly by H3T3A mutation can disrupt asymmetric histone inheritance. Further, the loss of asymmetric histone inheritance leads to a loss of asynchronous transcription re-initiation and asymmetric cell fate specification in dividing HBCs. We have included this data in Fig. 5F-J in the revised manuscript.

We are currently trying to generate an H3T3A-expressing mouse strain to study the *in vivo* role of asymmetric histone distribution, as this reviewer suggested.

Figure R11. (Figure 5F-J) H3T3A mutation disrupts both asymmetric histone and asymmetric p63 distribution in cultured HBCs. (A) Illustration showing the experimental procedure of H3T3A-Dendra2 lentivirus transfection and subsequent induction of H3.3-mScarlet expression in cultured HBCs. **(B)** An HBC colony with H3T3A-Dendra2 (green), H3.3-mScarlet (red), p63 staining (magenta) and Ki67 (blue). **(C)** Distribution pattern of H3.3-mScarlet and p63 in a telophase HBC. **(D)** Distribution pattern of Pol IIS2ph and p63 in a telophase HBC. **(E)** Distribution pattern of Pol IIS5ph and p63 in a telophase HBC. **(F)** Quantification of H3.3 ratio, Pol IIS2ph ratio, Pol IIS5ph ratio and p63 ratio in H3T3A-Dendra2+ and WT control telophase HBCs. H3T3A telophase HBCs: H3.3 ratio = 1.13 ± 0.12 (N=24); Pol IIS2ph ratio = 1.15 ± 0.02 (N = 22); Pol

IIS5ph ratio = 1.15 ± 0.02 (N = 23); p63 ratio = 1.12 ± 0.12 (N=24). Control telophase HBCs: H3.3 ratio = 1.35 ± 0.23 (N = 38); Pol IIS2ph ratio = 1.39 ± 0.04 (N = 45); Pol IIS5ph ratio = 1.33 ± 0.04 (N = 50); p63 ratio = 1.37 ± 0.04 (N = 45). All ratios: Mean \pm SEM. Mann-Whitney test, ** P<0.01, *** P<0.001, ****P < 0.0001. Scale bars: 20 μ m in (B) and 10 μ m (C-E).

Minor concerns:

Figure 2B, the DIC image does not correspond to the IF image. Could separate to avoid the misleading.

We appreciate the reviewer's suggestion and have separated the DIC and IF images accordingly.

The presentation of figures is not fluent, as Figure S2 is presented and then Figure S1C-E is cited.

In response to the reviewer's comment, we have reorganized Figure S1 and separated it into Supplementary Fig. 1 and Supplementary Fig. 3 to present the data sequentially.

Figure S4, The scale bars are not correct.

We thank the reviewer for pointing out this error and have corrected the legend of Supplementary Fig. 5 (original Fig. S4) in the revised manuscript.

Figure 5A is cited at a very early stage after Figure 2.

We assume the reviewer referred to the early mentioned "Fig. 5B". We have removed this sentence from the revised manuscript.

Figure S6, P-value is missing.

We have added all the p values in Supplementary Fig. 7 (original Fig. S6) in the revised manuscript.

Reviewer #5

We appreciate the reviewer's time and effort in evaluating our manuscript.

This is an interesting study. The authors show the biological significance of epigenetic histone inheritance and conservation of asymmetric histone inheritance in asymmetrically dividing stem cells. But I still have a few questions:

1. In the Tet-inducible H4-tag transgenic mouse model, H4-mScarlet is overexpressed or knock-in in situ? Some researchers do not accept the results from overexpressed histone tag, because their different characteristics with the endogenous histone.
2. The authors mentioned that histone H4, especially (H3.3-H4)₂ tetramers and SUV39H1 display asymmetric inheritance in HBCs. Does the (H3-H4) tetramers with specific modification, such as H3K27me₃, H3K9me₃, H3K27ac, display more significant asymmetric inheritance?
3. The authors mentioned that asymmetric mitotic spindle activity is required for asymmetric histone inheritance. Does DNA replication-coupled histone recycling participate in this process? eSPAN or SCAR-seq in the in-vitro cell model could explain this well.
4. It is somewhat difficult to understand that p63 high to maintain the quiescence of HBCs, Meanwhile, p63 high cells own higher H3.3-H4 tetramers and preferentially express the differentiation-related gene (scrNA-seq results). The author should explain this more clearly.
5. The authors disrupted the asymmetric histone inheritance by nocodazole treatment in mice and observed that the recovery in the OE thickness is slower than the control. How to exclude the effect of cell cycle arrest caused by nocodazole?

Minor:

1. The asymmetric of Pol IIS2ph distribution in the two daughter cells is not obvious.
2. The EU signal does not coincide with the Pol IIS2ph signal, please explain.